# Linking complex microbial interactions and dysbiosis through a disordered Lotka–Volterra model

**Jacopo Pasqualini[1†], Amos Maritan[1], Andrea Rinaldo[2,3], Sonia Facchin[4], Edoardo Vincenzo Savarino[4], Ada Altieri[5]\*, Samir Suweis[1,6]\***

[1]Dipartimento di Fisica "G. Galilei" e INFN sezione di Padova, Università di Padova, Padua, Italy; [2]Dipartimento di Ingegneria Civile, Edile e Ambientale (ICEA), Università di Padova, Padua, Italy; [3]École Polytechnique Fédérale de Lausanne, Lausanne, Switzerland; [4]Dipartimento di Scienze Chirurgiche, Oncologiche e Gastroenterologiche (DiSCOG), Università di Padova, Padua, Italy; [5]Laboratoire Matière et Systèmes Complexes (MSC), Université Paris Cité, CNRS, Paris, France; [6]Padova Neuroscience Center, University of Padova, Padua, Italy

**\*For correspondence:**
ada.altieri@u-paris.fr (AA);
samir.suweis@unipd.it (SS)

**Present address:** [†]University of Basel, Basel, Switzerland

**Competing interest:** The authors declare that no competing interests exist.

## eLife Assessment

This **important** study shows how the relative importance of inter-species interactions in microbiomes can be inferred from empirical species abundance data. The methods based on statistical physics of disordered systems are **compelling** and rigorous, and allow for distinguishing healthy and non-healthy human gut microbiomes via differences in their inter-species interaction patterns. This work should be of broad interest to researchers in microbial ecology and theoretical biophysics.

**Abstract** The rapid advancement of environmental sequencing technologies, such as metagenomics, has significantly enhanced our ability to study microbial communities. The eubiotic composition of these communities is crucial for maintaining ecological functions and host health. Species diversity is only one facet of a healthy community's organization; together with abundance distributions and interaction structures, it shapes reproducible macroecological states, that is, joint statistical fingerprints that summarize whole-community behavior. Despite recent developments, a theoretical framework connecting empirical data with ecosystem modeling is still in its infancy, particularly in the context of disordered systems. Here, we present a novel framework that couples statistical physics tools for disordered systems with metagenomic data, explicitly linking diversity, interactions, and stability to define and compare these macroecological states. By employing the generalized Lotka–Volterra model with random interactions, we reveal two different emergent patterns of species interaction networks and species abundance distributions for healthy and diseased microbiomes. On the one hand, healthy microbiomes have similar community structures across individuals, characterized by strong species interactions and abundance diversity consistent with neutral stochastic fluctuations. On the other hand, diseased microbiomes show greater variability driven by deterministic factors, thus resulting in less ecologically stable and more divergent communities. Our findings suggest the potential of disordered system theory to characterize microbiomes and to capture the role of ecological interactions on stability and functioning.

## Introduction

Microbial communities are a fundamental reservoir of ecological functions and biological diversity. They are relevant for any environmental and host-associated ecosystem, ranging from soil to the human gut. In particular, the human gut microbiome, by interacting with the host's metabolism and immune system (*Levy et al., 2016*; *Barroso-Batista et al., 2015*; *Barreto and Gordo, 2023*), is a key regulator of human health. Dysbiosis is defined as an alteration in the composition of healthy microbiomes associated with the gastrointestinal tract (*Lloyd-Price et al., 2019*; *Das and Nair, 2019*; *Nishida et al., 2018*).

Recently, stochastic non-interacting neutral (*Zeng et al., 2015*; *Venkataraman et al., 2015*; *Sala et al., 2016*; *Sieber et al., 2019*; *Descheemaeker and de Buyl, 2020*; *Grilli, 2020*; *Zaoli and Grilli, 2022*), and logistic models have been successfully used to describe empirical patterns such as species abundance distribution (SAD) and species presence/absence statistics in different types of communities. However, focusing on single-species properties (*Azaele et al., 2016*; *Grilli, 2020*) fails to characterize altered states of microbiome (*Seppi et al., 2023*; *Pasqualini et al., 2024*). In particular, recent studies indicate that gut dysbiosis is associated with shifts in microbial species interaction patterns. Network-level properties of species interactions – such as the balance of positive and negative interactions, average interaction strength, and connectivity of the inferred interaction matrix – have been shown to systematically differ between healthy and dysbiotic gut communities (see, for instance, *Bashan et al., 2016*; *Seppi et al., 2023*). Moreover, pairwise species interactions have been quantified in simplified in vitro consortia (*Kehe et al., 2021*; *Venturelli et al., 2018*).

Motivated by these findings, several approaches have been proposed in recent years to infer microbial interactions (*Faust and Raes, 2012*; *Xiao et al., 2017*; *Camacho-Mateu et al., 2024*). However, such inference protocols remain very challenging and problematic in many respects (*Faisal et al., 2010*; *Angulo et al., 2017*; *Tu et al., 2019*; *Armitage and Jones, 2019*; *Holt, 2020*): (a) the very high dimension of typical microbiome datasets and the lack of longitudinal (long-term) experiments; (b) the time-dependent nature of microbial species interactions (*Pacciani-Mori et al., 2021*); (c) abiotic factors such as resources, temperature, and pH can introduce environmental filtering effects (*Sireci et al., 2022*) and induce effective interactions; and (d) experimental and technical challenges that introduce sampling effects, false positive species (*Tovo et al., 2020*), spurious correlation, and bias in the data (*Weiss et al., 2016*; *Gloor et al., 2017*; *Dohlman and Shen, 2019*). Therefore, even in a scenario where metagenomic samples are noise- and bias-free, reconstructing species interaction networks of the underlying microbial dynamics remains a hard task.

An alternative theoretical approach was pioneered by *May, 1972*, proposing to use random matrices to model species interaction networks, that is, each entry of the adjacency matrix is extracted at random from a given distribution. Given the impossibility of empirically measuring the interaction strengths, the advantage of such an approach is the reduction from $\sim S^2$ (if $S$ is the number of species) to just a few parameters (e.g., 2) used to parameterize the distribution (*Allesina and Tang, 2012*).

## Insights from disordered systems theory

Although previous studies have largely focused on neutral theory and logistic models – including variants like the Stochastic Logistic Model that account for environmental fluctuations – they often neglect inter-species interactions, limiting their ability to reproduce macroecological patterns at a global scale. Moreover, higher-order correlations among species are known to generate non-trivial effects, such as the emergence of persistent fluctuations or chaotic dynamics due to the non-reciprocity of interactions.

In recent years, increasing attention has been devoted to studying population dynamics through the generalized Lotka–Volterra (gLV) equations, employing disordered systems techniques (also known as *glassy*), such as replica and cavity methods and Dynamical Mean-Field Theory approaches, originally developed in the context of statistical physics (*Bunin, 2017*; *Galla, 2018*; *Biroli et al., 2018*; *Altieri et al., 2021*; *Lorenzana and Altieri, 2022*). Indeed, due to the inherently high dimensionality of microbial datasets, random matrix theory and methods from disordered systems turn out to be particularly well suited. A striking feature of their application to ecological dynamics is that the resulting properties and dynamical regimes do not depend on species-specific details, provided that species are statistically equivalent under relabeling or time-averaging. Phase diagrams can thus be established in terms of a few effective parameters (e.g., the mean $\mu$ and the variance $\sigma^2$ of the random

interactions strengths; *Barbier et al., 2018*). However, despite a range of very interesting results, this approach has remained confined mainly to purely theoretical domains (but see *Hatton et al., 2024*) and has only recently been used to give an interpretation of controlled experiments with synthetic microbial communities (*Hu et al., 2022*).

## Results

We now provide a *proof of concept* of the applicability of a high-dimensional disordered setting to human microbiomes. Specifically, in the following section, we introduce the disordered generalized Lotka–Volterra (dgLV) model, infer its parameters from healthy and unhealthy cohorts of gut microbiomes, and test whether the resulting interaction patterns and stability metrics discriminate healthy from diseased macroecological states. Finally, we will introduce quantitative metrics to define microbiome stability and estimate the contribution of distinct ecological forces to the dynamics.

### dgLV model

The dgLV model describes the time evolution of the concentration abundances of a local pool of $S$ interacting species, that is,

$$\frac{dN_i}{dt} = N_i \left[ \rho_i(K_i - N_i) - \sum_{j,(j \neq i)} \alpha_{ij} N_j \right] + \sqrt{N_i} \eta_i(t) + \lambda \,, \tag{1}$$

where $N_i$ is the population of species $i$ th, $K_i$ is its carrying capacity, and $r_i$ the growth rate, where $\rho_i = \frac{r_i}{K_i}$ is a constant, which we will assume will not depend on the species, that is, $\rho_i = \rho$. The coefficients $\alpha_{ij}$ are i.i.d. random variables with $\mathbb{E}[\alpha_{ij}] = \mu/S$ and $\mathbb{E}[\alpha_{ij}]^2 - \mathbb{E}^2[\alpha_{ij}] = \sigma^2/S$. We incorporate a demographic noise term with variance defined by $\langle \eta_i(t)\eta_j(t') \rangle = 2T\delta_{ij}\delta(t - t')$. Precisely, $\delta_{ij}$ is the Kronecker delta for species indices, and $T$ sets the scale of noise intensity. This framework captures demographic fluctuations within a continuous description (*Domokos and Scheuring, 2004*; *Rogers et al., 2012*), with the noise amplitude $T$ being inversely proportional to the total population size (*Altieri et al., 2021*). For notational purposes, we also introduce its inverse $\beta = T^{-1}$. Then, we include a species-independent immigration rate $\lambda$, which will be treated as a reflecting wall mathematically regularizing the problem.

By requiring that the interactions are symmetric, that is, $\alpha_{ij} = \alpha_{ji}$, we can map this problem to an equilibrium thermodynamic one that is exactly solvable. As shown in the Appendix, section S1, it is possible to justify this symmetric assumption and a linear dependence between the growth rate and the carrying capacity, namely $r_i = \rho K_i$, by considering the quasi-stationary approximation of the MacArthur consumer–resource model. In the following, we set $\rho = 1$ (following also *Biroli et al., 2018*). The presence of a noise term in *Equation 1* allows us to write the Fokker–Planck equation of the system and study its stationary solution (see *Altieri et al., 2021*; *Altieri and Biroli, 2022* for a detailed derivation in a similar Hamiltonian formalism). Here, we adopt the Itô prescription for the stochastic dynamics in such a way as to prevent species resurgence by noise.

The replica formalism, a well-known technique in disordered systems, comes into play and allows us to derive a non-interacting Hamiltonian corresponding to the dgLV symmetric interactions $\alpha_{ij}$ (see Methods and Appendix, Section S2, for a complete derivation). In simple words, instead of trying to solve the problem for one random setup, the formalism considers many replicas of the system and averages their behaviors. This approach helps to smooth out the randomness and reveals the typical behavior of the system. We also assume a single equilibrium scenario, known as *replica-symmetric* (RS) regime. Although the SADs of empirical microbial communities display fat tails, a feature more compatible with the multi-attractors phase of the asymmetric 1RSB (*Mallmin et al., 2024*) case or with the gLV with time-dependent disordered interactions (*Suweis et al., 2024*) (*annealed* version), we consider this simplification as it is the regime in which we can obtain explicit analytical relations between the model parameters and the data and where model inversion is feasible. By employing a cavity argument (*Mezard and Montanari, 2009*; *Altieri and Baity-Jesi, 2024*), one can indeed analytically derive the SAD of the model (see Methods and Appendix, Section S4):

$$p(N|\zeta) \propto N^{\nu-1} \exp\left\{ -\beta\left( \frac{m}{2}N^2 - \zeta N \right) \right\}, \tag{2}$$

where the auxiliary variable $\zeta = K - \mu h + \sqrt{q_0}\sigma z$ takes the disorder interactions into account; $z$ is a standardized Gaussian variable, $\nu = \beta\lambda > 0$. We introduce, respectively, the mean abundance $h = \overline{\langle N \rangle}$, the self-overlap $q_d = \overline{\langle N^2 \rangle}$, and the overlap $q_0 = \overline{\langle N \rangle^2}$, which we will collectively refer to as *order parameters* in the following sections. Specifically, $q_0$ measures the similarity between two different configurations at stationarity of the system with two different disorder realizations, while $q_d$ measures the similarity of two stationary configurations generated with the same disorder realization. Finally, we introduce a constant that, inspired by the Field Theory jargon, we dub as mass $m = 1 - \beta\sigma^2(q_d - q_0) > 0$ for the theory to be consistent (*Biroli et al., 2018*).

The RS ansatz can also be characterized by its stability to external perturbations, such as the external temperature, the immigration rate, or the interaction heterogeneity. To investigate the stability of the RS phase, we consider the Hessian matrix of the theory by performing a harmonic expansion of the replicated free energy, as originally pointed out in *de Almeida and Thouless, 1978*. When the leading eigenvalue of the Hessian matrix, the so-called *replicon mode*, goes below zero, unstable equilibria appear: either the system moves toward a one-step replica symmetric breaking (1RSB) phase with multiple locally stable minima, or it develops a marginal full-RSB phase, leading to a hierarchical structure of states and therefore to an extremely rough landscape (also known as *Gardner amorphous-like phase*). In other words, the RS ansatz no longer represents a thermodynamically stable phase in the latter cases.

Based on standard calculations in disordered systems, the *replicon* mode $\mathcal{R}$ can be analytically evaluated and reads:

$$\mathcal{R} = (\beta\sigma)^2\left(1 - \sigma^2\overline{\langle\left(\frac{\partial N}{\partial\xi}\right)^2\rangle}\right) = (\beta\sigma)^2\left(1 - (\beta\sigma)^2\overline{(\langle N^2 \rangle - \langle N \rangle^2)^2}\right),\tag{3}$$

where $\frac{\partial N}{\partial\xi}$ is the species response to an external perturbation, keeping track of non-extinct species only. (The replicon eigenvalue refers to a particular type of fluctuation around the saddle-point (mean-field) solution in the replica framework. When diagonalizing the Hessian matrix of the replicated free energy, fluctuations split into three sectors: longitudinal, anomalous, and replicon. The replicon mode

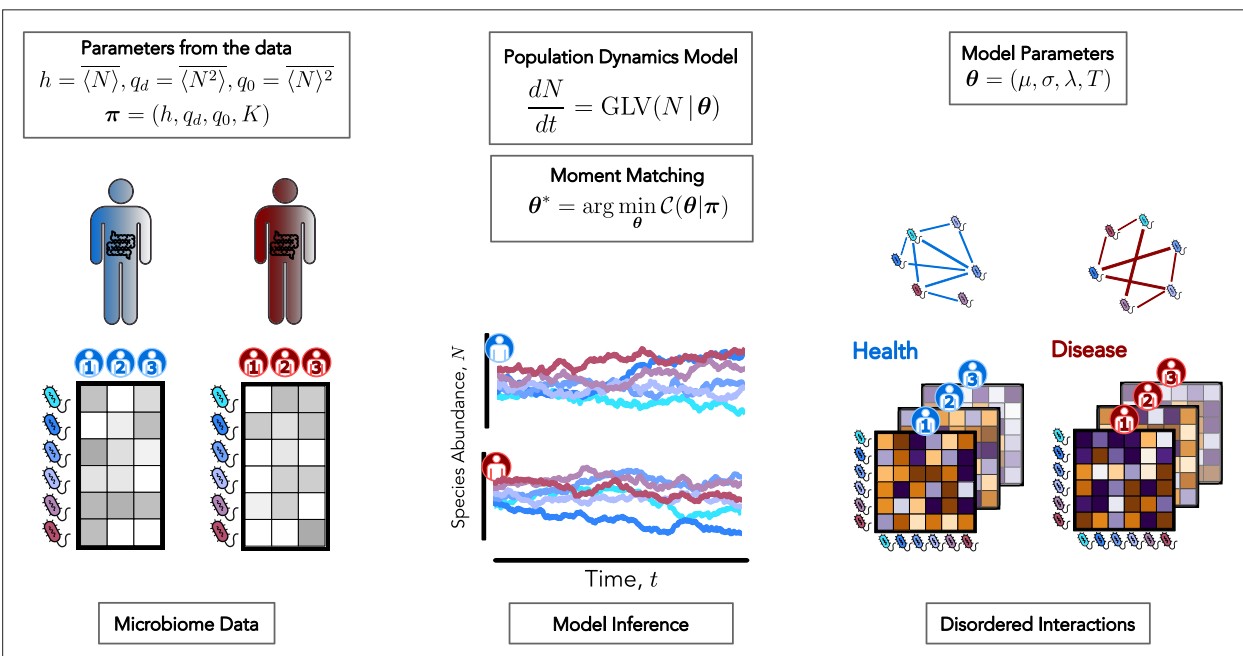

**Figure 1.** The inference protocol of the dgLV generative model is performed by a moment matching optimization procedure. We aim to infer the free parameters $\theta = (\mu, \sigma, T, \lambda)$ – shown on the right – so that to match the mean abundance, higher-order correlations between species abundances, and the average carrying capacity for the two cohorts, that is, $(h, q_0, q_d, K)$ – on the left. This procedure enables us to extract relevant information about the ecological dynamics from cross-sectional data of healthy (blue) and diseased (red) microbiomes, which are treated as independent disordered realizations.

is the most sensitive to criticality signaling – by its vanishing trend – the emergence of many nearly degenerate states. It essentially describes how 'soft' the system is to microscopic rearrangements in configuration space). For more details, see S3 in the Appendix.

## Data through the glass

We now consider cross-sectionally sampled gut microbiomes of two different cohorts: one of healthy and another of diseased individuals. Here we focus on chronic inflammation syndromes, but, in principle, what we present holds for groups of phenotypically distinct populations. We can idealize the samples of each group as generated from the stationary distribution of the dgLV *Equation 1*, with *different* realizations of the disorder $\boldsymbol{\alpha}$, but with shared $\mu$ and $\sigma$ (see Materials and methods). The approach we propose considers all samples in the same group as coming from the same statistical ensemble. On the other hand, it distinguishes ecological regimes characterized by distinct phenotypes through different statistics (e.g., dissimilar $\mu$ and $\sigma$) of the random species interactions $\boldsymbol{\alpha}$. We aim to test the hypothesis that each cohort will be described by a different set of ecological parameters, precisely with $\boldsymbol{\theta} = (\mu, \sigma, T, \lambda)$.

To calculate the order parameters $(h, q_d, q_0, K)$ from the data, we thus need to specify how we perform the ensemble and the disorder averages empirically (see *Figure 1*). Since time series data are rarely available, we rely on an effective mean-field description and estimate the ensemble average by averaging over species (that is, different species are realizations of the same underlying stochastic process *Azaele et al., 2016*, known as neutral hypothesis), that is, $\langle \cdot \rangle \sim \frac{1}{S} \sum_{species} \cdot$. Then, we assume that the average over the disorder can be computed as a sample average. In other words, within a given phenotype (healthy/unhealthy), each measured microbiome configuration is a sample from the stationary distribution of the dgLV model, with a given realization of the disorder, that is, $\overline{\cdot} \sim \frac{1}{R} \sum_{samples} \cdot$, where $R$ is the number of samples and typically $R \gg 1$. Moreover, due to our limited knowledge of the fine details governing the interactions in each microbiome, it is reasonable to assume that all communities with a given macrostate – a point in the $(h, q_0, q_d, K)$ space – experience the same demographic noise $T$, immigration rate $\lambda$, and disorder parameters $\mu$ and $\sigma$. Eventually, we can compute the order parameters from the data as:

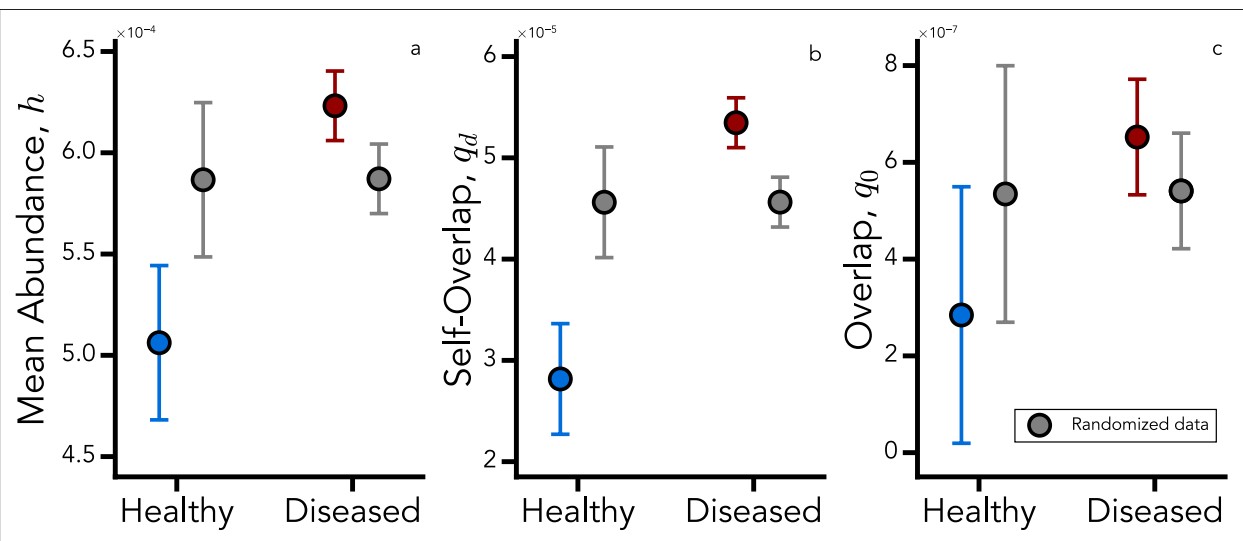

**Figure 2.** Order parameters inferred from the data using *Equation 4*. (**a**) shows $h$, (**b**) $q_d$, and (**c**) $q_0$ in healthy (blue) and unhealthy (red) cohorts. Circles denote the mean value of each order parameter, and error bars indicate the standard deviation across bootstrap realizations. Bootstrap estimates were obtained from 5000 iterations, each retaining 90% of samples within each cohort. Gray symbols show the corresponding null-model values obtained by randomizing cohort labels prior to estimation.

$$h = \overline{\langle N \rangle} = \frac{1}{R} \sum_{a=1}^{R} \left( \frac{1}{S_a} \sum_{j=1}^{S} N_{j,a} \right),$$

$$q_d = \overline{\langle N^2 \rangle} = \frac{1}{R} \sum_{a=1}^{R} \left( \frac{1}{S_a} \sum_{j=1}^{S} N_{j,a}^2 \right), \qquad (4)$$

$$q_0 = \overline{\langle N \rangle}^2 = \frac{1}{R} \sum_{a=1}^{R} \left( \frac{1}{S_a} \sum_{j=1}^{S} N_{j,a} \right)^2,$$

where $N_{j,a}$ represents the population density of species $j$ in sample $a$.

To evaluate the carrying capacities $K_i$ from the data, as done in most of the works using dgLV equations (**Bunin, 2017**; **Biroli et al., 2018**; **Altieri et al., 2021**; **Suweis et al., 2024**; **Mallmin et al., 2024**), we assume that in each cohort all species are characterized by the same carrying capacity $K_i = K$. Here, we define $K$ as the average of the maximum relative abundances of species across the available samples for each cohort, so that $K = \frac{1}{S} \sum_{j=1}^{S} \max_a N_{j,a}$.

Note also that, from metagenomics, we only have access to compositional data for species abundances (**Pasqualini et al., 2024**). This is crucial to properly treat different samples and end up with a consistent analysis. Moreover, this approach allows us to give an ecological interpretation of some of the order parameters. In particular, because of the compositionality of the data, we have that $h = \overline{S^{-1}}$ and $q_0 = \overline{S^{-2}}$, while $q_d$ provides information on pairwise products between species abundances within each replica. **Figure 2** shows the values of such order parameters between healthy and unhealthy cohorts for two randomized instances. In the data, we observe a systematic difference between healthy and unhealthy cohorts, pointing to a higher average local diversity in healthy samples (panels a, c), as also observed in **Pasqualini et al., 2024**. Panel b highlights a higher $q_d$ in unhealthy patients, a signature of the weakening species interactions in those samples. We will further investigate this aspect by inferring the species interactions in both cohorts.

From the modeling point of view, it can be rigorously shown that constraining the abundances to be normalized to one corresponds to introducing a Lagrange multiplier in the expression of the free energy. However, this only acts on the linear term, contributing to shifting the mean of the random interactions, $\mu$, but does not affect the heterogeneity $\sigma$, and therefore the phase diagram overall. For more details, we refer the interested reader to the Appendix (end of Section S2) along with (**Altieri**

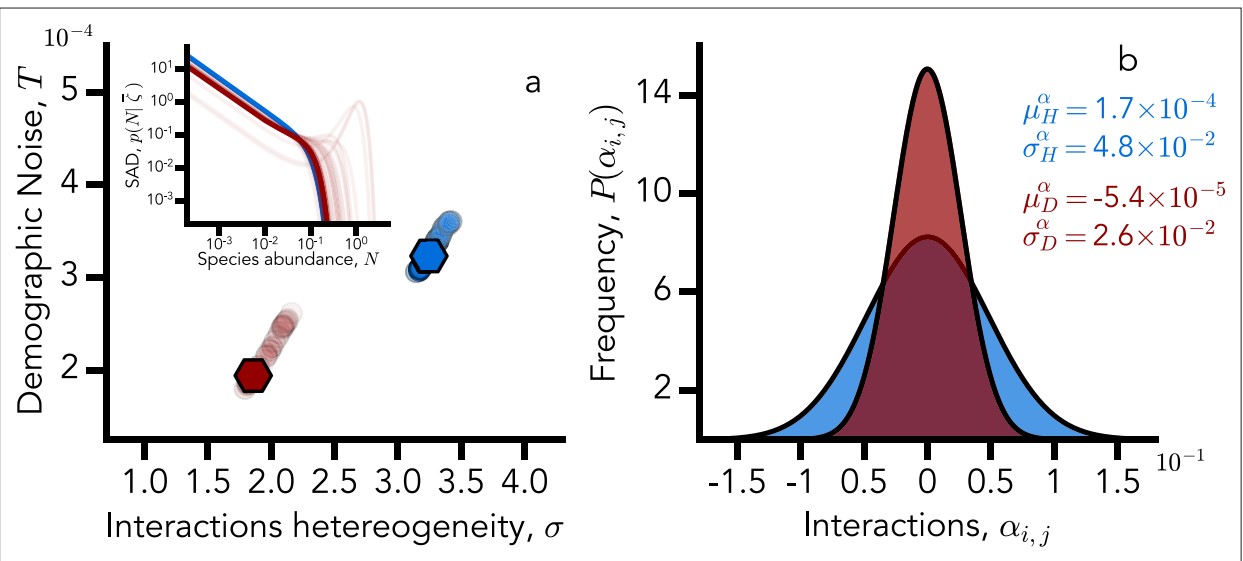

**Figure 3.** Distinct ecological organization in healthy vs diseased microbiomes. (**a**) Inferred $T$ (demographic noise strength) and $\sigma$ (interactions heterogeneity) for healthy (blue) and diseased (red) microbiomes are clustered. Darker dots correspond to better solutions (i.e., solutions with a lower value of the cost function $\mathcal{C}$), while the two points with hexagonal markers correspond to the best two (healthy and diseased, respectively) solutions. In the first panel inset, we also show (in log–log scale) the species abundance distributions (SADs) corresponding to each solution. To have a more concise representation, we present each SAD fixing the disorder to its average $\overline{\zeta} = K - \mu h$. (**b**) The probability density function of the inferred interactions $\alpha_{i,j}$ for healthy (blue) and diseased (red) microbiomes. Dysbiosis reduces the heterogeneity of the interaction strengths. The quantities reported in the legend are the average and standard deviation of $\alpha_{i,j}$. They are calculated as $\mu_X^{\alpha} = \mu_X/S_X$ and $\sigma_X^{\alpha} = \sigma_X/\sqrt{S_X}$, where $S_X$ is the species pool size, estimated as the set of all observed species in a dataset, $X$ can denote healthy ($H$) or diseased ($D$) individuals.

*et al., 2021*). In *Altieri et al., 2021*, the authors also showed how a global normalizing constraint on species abundances reflects in a one-to-one mapping with the random replicant model (*Biscari and Parisi, 1995*). (The replicator equations originally introduced by *Diederich and Opper, 1989* and recast within the replica formalism *Biscari and Parisi, 1995*; *Altieri et al., 2021* describe the dynamics of an ensemble of replicants that evolve via random couplings).

## Species interaction patterns characterize the state of microbiomes

We thus collect all the parameters estimated from the data in a vector $\boldsymbol{\pi} = (h, q_d, q_0, K)$. As we will better detail in the Methods, we develop a moment matching inference algorithm to infer the model parameters $\boldsymbol{\theta}$, as depicted in *Figure 1*. The idea of the method is to introduce a cost function $\mathcal{C}(\boldsymbol{\theta}|\boldsymbol{\pi})$, representing a total relative error for some self-consistent equations. If the parameters $\boldsymbol{\theta}$ are such that the right part of the self-consistent equation equals the left part, the problem is considered solved. Because the landscape associated with this cost function presents several minima, we perform multiple optimization procedures to collect an ensemble of possible solutions, from which we retain the top 30. First, we find that different solutions $\boldsymbol{\theta}^*$ of the optimization problem provide ecological insights into the underlying microbiome populations.

As originally predicted in *Altieri et al., 2021*, among all the parameters that define *Equation 1*, the only ones relevant for reproducing the theoretical phase diagram are the amplitude of demographic noise and the heterogeneity of interactions. The mean interaction strength, provided it is sufficiently positive, does not play a significant role. This prediction is fully confirmed by the inference procedure applied to the two microbiome datasets, allowing us to identify a universal signature that distinguishes healthy from unhealthy states. *Figure 3a* shows, indeed, that inferred noise ($T$) and interaction heterogeneity strength ($\sigma$) for healthy and diseased microbiomes are clustered in the two-dimensional plane.

In particular, the SAD for the healthy cohort is robust among the different solutions of the inference procedure, as depicted by the superposition of the different curves in the inset of *Figure 3a*. On the other hand, SADs inferred from unhealthy patients have high sensitivity to different solutions. In particular, some of them display a mode for high-abundance species (light red lines in *Figure 3a*), a signature of dominant strain in the gut. Consistently, the distribution of the interactions $P(\alpha_{i,j})$ generated through the inferred parameters $\mu$ and $\sigma$ is different between healthy and diseased cohorts, giving a

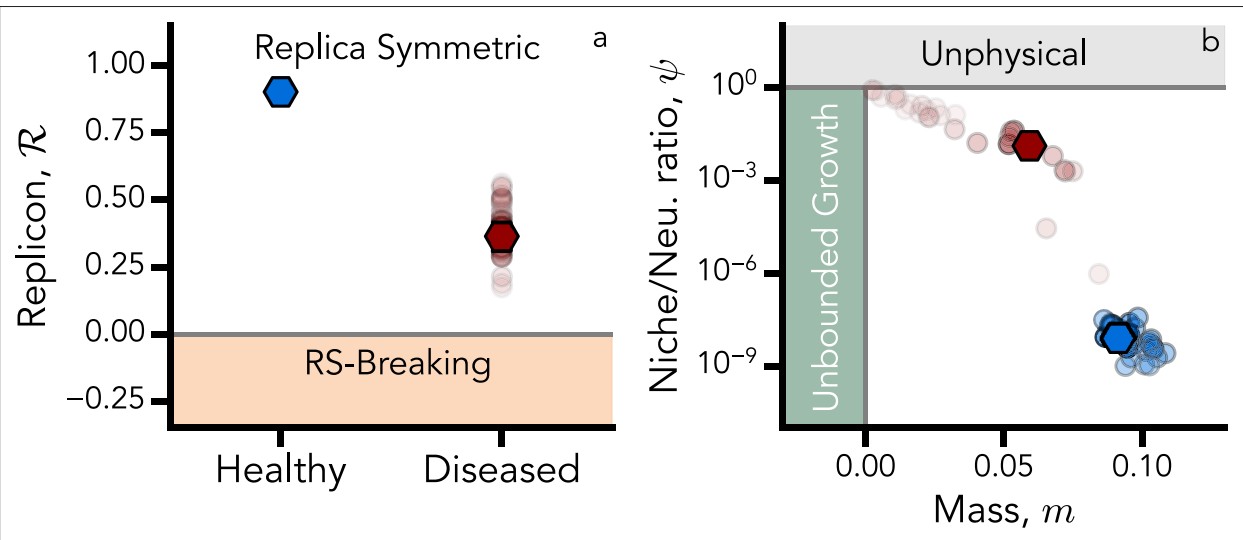

**Figure 4.** Stability of healthy vs diseased microbiomes. (**a**) The replicon eigenvalue corresponding to each solution of our optimization procedure (shaded dots). The solid hexagon represents the replicon corresponding to the best solutions that minimize the error in predicting the order parameters of the theory (minimum $\mathcal{C}$). The two investigated microbiome phenotypes (healthy in blue, diseased in red) are significantly different. In particular, diseased microbiomes are closer to the marginal stability of replica-symmetric ansatz (gray horizontal line). (**b**) Solutions of the moment-matching objective function are shown as a function of $\psi$ and $m$, which in turn depend on the species abundance distribution (SAD) parameters (see main text). Healthy (blue) and diseased (red) microbiomes appear to be clustered. Therefore, distinct ecological organization scenarios (strong neutrality/emergent neutrality) take place. Darker dots correspond to solutions with lower values of the cost function, while hexagonal markers correspond to the two best solutions.

distinct pattern of interactions (see *Figure 3b*), a result that is compatible with that found by *Bashan et al., 2016*. Remarkably, we find that dysbiosis reduces the heterogeneity of interaction strengths, a result also observed when taking correlations as a proxy for interactions (*Seppi et al., 2023*).

We then assess how close the inferred $\sigma$ and $\beta = 1/T$ (a.k.a. inverse temperature in a statistical physics approach) are to the critical RSB line of the dgLV ($\mathcal{R} = 0$), evaluated by keeping all the other parameters constant (see Methods). We find again that the replicon values $\mathcal{R}$ corresponding to each solution of our optimization protocol are significantly different for the two investigated microbiome phenotypes (see *Figure 4a*). In particular, diseased microbiomes are closer to marginal stability within the RS ansatz (*Altieri et al., 2021*; *Mézard et al., 1987*; *de Almeida and Thouless, 1978*). Furthermore, by investigating the shape of the SAD given by *Equation 2*, we can estimate the ratio between niche (represented by species interaction) and neutral (represented by birth/death and immigration) ecological forces, which can be captured by the quantity $\psi$ (*Wu et al., 2021*). It detects the emergence of peaks in the SAD as a hallmark of niche processes (see Appendix 2).

Inspired by field-theory arguments (see Methods and Appendix, Section S2), we can call the *mass* of the theory the $m$ parameter, as defined above in *Equation 2*. In classical and quantum field theory, the particle–particle interaction embedded in the quadratic term is typically referred to as a mass source. In our context, $m = 1 - \beta\sigma^2(q_d - q_0)$ captures quadratic fluctuations of species abundances, as also appearing in the expression of the leading eigenvalue of the stability matrix. When $m \to 0$, the analytical order parameters diverge and the system enters the unphysical regime of unbounded growth. As such, the *mass* term can be considered a complementary stability measure, capable of capturing the transition to the unbounded growth regime.

In the model, two kinds of effects compete to shape the community structure. On the one hand, we have niche effects, encoded in disordered interactions and thus tracked by the parameters $\mu$, $\sigma$, and $K$. Their overall effect is selective and tends to concentrate the SAD around the typical abundance value. On the other hand, we have neutral effects encoded in the stochastic dynamics and immigration, governing the low-abundance regime of the SAD. When the demographic noise amplitude is stronger than immigration ($\nu < 1$, as in our case), the SAD exhibits a low-abundance integrable divergence. In the opposite scenario, for $\nu > 1$, there is no divergence, and the SAD is modal. Since interactions are random, the probability of observing an internal mode can be estimated as the fraction of SADs realizations having non-trivial solutions to the stationary point equation. Such a quantity, dubbed as the niche–neutral ratio, can be analytically evaluated:

$$\psi = \frac{1}{2}\mathrm{Erfc}\left(\frac{\zeta^* + \overline{\zeta}}{\sqrt{2}\sigma_\zeta}\right) + \frac{1}{2}\mathrm{Erfc}\left(\frac{\zeta^* - \overline{\zeta}}{\sqrt{2}\sigma_\zeta}\right) \, , \tag{5}$$

where $\zeta^* = \sqrt{\frac{4(1-\nu)m}{\beta}}$ and $\overline{\zeta} = K - \mu h$. When $\psi \approx 1$, niche and neutral forces give comparable contributions to the dynamics, as both low-abundance divergence and a finite abundance mode coexist in the SAD. Finally, if the typical abundance diverges, we enter the unbounded growth phase, which means that the mass $m$ and the niche–neutral ratio $\psi$ are not independent, as suggested by the analytical expression for $\psi$. For an exhaustive derivation of this result, see Appendix 2. With the obtained model parameters, we are able to evaluate $m$ and $\psi$ for healthy and diseased microbiomes. Also, in this case, healthy and diseased microbiomes are visibly clustered, as shown in *Figure 4*. Unhealthy microbiomes turn out to be closer to the unbounded growth phase, and the niche–neutral ratio is larger by five orders of magnitude than the healthy case $\psi_D \approx 10^5 \psi_H$. This leads us to argue that selective pressure is way larger in diseased states, while in the healthy one, birth and death effects are the key drivers of the dynamics. These results are also confirmed by the SAD shapes in the inset of *Figure 3* (panel a).

In summary, in the Results section, we show that (i) the inference pipeline robustly recovers demographic noise and interaction heterogeneity by calculating $h$, $q_0$, and $q_d$ from the data; and (ii) these parameters cluster according to health status, with diseased microbiomes lying closer to the replica-symmetry-breaking threshold, indicating reduced ecological resilience.

## Discussion

In our exploration of the gut microbiome through the lens of disordered systems and random matrix theory, we have proposed a connection between the theoretical framework of disordered systems and practical analyses of environmental microbiome data. In particular, we have characterized healthy and unhealthy gut microbiomes using the dgLV model for population dynamics. We now interpret our theoretical findings in a biological context, contrast them with previous work, and outline limitations and future directions.

The first major result of our work suggests a different role for the various ecological forces shaping the human gut microbial community. In this sense, the niche–neutral ratio $\psi$, highlights the different roles of interactions in healthy and diseased microbiomes. In the healthy case, neutral forces, such as random birth and death of individuals, characterize the dynamics, making configurations corresponding to this state alike. On the contrary, in diseased microbiomes, disordered, sample-specific interactions are the dominant ecological force, making individual realizations differ significantly from one another. An ecological interpretation of our findings suggests that healthy microbiomes are governed primarily by demographic stochasticity, reflecting a quasi-neutral regime characterized by similar community structures across individuals. Conversely, microbiomes from diseased patients exhibit significantly greater variability, suggesting that deterministic ecological factors – such as weakened species interactions – override neutrality, leading to structural instability and distinct microbial compositions. This observation aligns with the 'Anna Karenina principle' (*Pasqualini et al., 2024*; *Ma, 2020*) holding for gut microbiomes, which can be phrased as: 'All healthy gut microbiomes are alike; each unhealthy gut microbiome is unhealthy in its own way'. Supporting this interpretation, our analysis of the replicon eigenvalue $\mathcal{R}$ shows that the healthy state is associated with pronounced stability to external perturbations. The unhealthy state, instead, being closer to the RSB line, exhibits diminished stability and, consequently, reduced robustness against external perturbations.

Our study also sheds light on the role of demographic noise within the context of the dgLV model. In the limit $T \rightarrow 0$, the SAD transitions to a truncated Gaussian, with a prominent peak at zero reflecting the fraction of extinct species. In contrast to a zero-noise scenario, where species extinctions are observed, the inclusion of demographic noise in the dgLV model suggests a picture where no species goes extinct, supporting the *everything is everywhere* hypothesis (*Grilli, 2020*; *Pigani et al., 2024*). In other words, within this framework, zeros in the data are due to sampling effects and not due to local species extinctions (*Grilli, 2020*; *Pasqualini et al., 2024*).

A notable limitation of our study lies in the discrepancy between the empirical SAD observed in the data and the theoretical distribution predicted by the quenched dgLV model (*Suweis et al., 2024*). While empirical data showcase a diverse range of species abundance, following a power-law distribution, the model predictions tend to exhibit exponential decay in SAD tails. This mismatch underscores the need for further refinement of the model to accurately capture the nuanced patterns observed in real-world data. For example, it has recently been proposed that introducing an annealed disorder (unlike the *quenched* approximation, the *annealed* version assumes that random couplings are not fixed but rather fluctuate over time, with their covariance governed by independent Ornstein–Uhlenbeck processes) can generate SADs that more closely resemble the empirical ones (*Suweis et al., 2024*). Another possibility is to set the dgLV model parameters in such a way as to reproduce the multi-attractor phase. In fact, in this region, the SADs display a more heterogeneous shape (*Arnoulx de Pirey and Bunin, 2024*). We plan to explore this follow-up direction by combining 1RSB computations in the multiple-attractor phase with Dynamical Mean-Field Theory analysis for the asymmetric interaction case. This latter approach is particularly well suited for studying inherently non-equilibrium dynamics and for extending the framework to systems subject to environmental fluctuations in addition to demographic noise.

Another related limitation is the challenge of generating species abundance samples from the dgLV model that mirrors the statistical properties of the observed empirical data. In our current framework, each microbiome sample could be extracted from $p(N|\zeta)$, where $\zeta$ is a realization of the disorder. However, $p(N|\zeta)$ near $N \rightarrow 0$ presents a power-law exponent $\nu - 1$, with $\nu_{H,D} \approx 10^{-3}$. This results in numerical instabilities and dominates the sampling process, posing difficulties in generating representative synthetic samples. Moreover, while microbiome data are inherently compositional (*Pasqualini et al., 2024*), the dgLV model species populations $N_i$ are positive real numbers. However, as already noted, it can be shown that introducing a normalization $\tilde{N}_i \rightarrow N_i / \sum_j N_j$ in such equations does not

change the structure of the proposed solutions (*Altieri et al., 2021*) and therefore should not affect the conclusions of our work.

In conclusion, our work proposes a bridge between theory and data, particularly in refining the theoretical models to better align with empirical observations and in exploring the nuances of SADs within the microbiome context. Moreover, the integration of other forms of environmental variability and species-specific traits could provide a more holistic view of ecological dynamics, as also proposed by the *One Health-One Microbiome* framework (*Tomasulo et al., 2024*).

Overall, our study builds a quantitative link between metagenomic data and the disordered gLV framework, revealing how dysbiosis alters gut species interaction networks. By doing so, it lays the groundwork for more advanced, mechanistically informed models to better interpret and ultimately manage complex microbial ecosystems.

# Materials and methods

## Microbiome dataset and code

We have selected gut microbiome data from three studies (*Franzosa et al., 2019*; *Lloyd-Price et al., 2019*; *Mars et al., 2020*), focusing on inflammatory syndromes of the gastrointestinal tract (Crohn's disease, ulcerative colitis, and irritable bowel syndromes). Considering all the available metadata, we have selected the patients less affected by possible perturbing factors, such as drugs. Finally, our dataset consists of $R_{\text{Healthy}} = 91$ shotgun metagenomic samples from healthy control individuals and $R_{\text{Diseased}} = 202$ shotgun metagenomic samples. All metagenomic preprocessing and reads classification are extensively described in *Pasqualini et al., 2024*. Species abundance profiles from metagenomic data and the parameter values obtained from the moment matching inference are available at Zenodo.

All the scripts implementing the moment matching inference and the Jupyter notebooks to generate the figures are available at GitHub (copy archived at *Pasqualini, 2026*).

## Free-energy landscape exploration: replica formalism

In the case of symmetric interactions – corresponding to conservative forces in the dynamics – a one-to-one mapping between the multi-species dynamics and a thermodynamic formalism can be safely worked out. The first step consists of writing the Fokker–Planck equation in the presence of a white Gaussian noise defined by a zero mean and a variance of amplitude $2T$. All technical details, leveraging a Fokker–Planck derivation, can be found in *Altieri and Biroli, 2022* for a similar pairwise interacting model, but a different self-regulation term accounting for non-logistic behavior (*Equation 1*).

Once the (quenched) disordered Hamiltonian of the model is obtained, we can resort to techniques known in statistical physics of disordered systems, such as the replica and cavity methods (*Mezard and Montanari, 2009*; *Mézard et al., 1987*; *Zamponi, 2010*). The replica trick, in particular, allows us to handle disordered quantities, such as the free energy and the partition function, which would be otherwise unaffordable (see Appendix).

We summarize the main findings here along with the expressions of the (RS) order parameters of the model, $(h, q_d, q_0)$. The three expressions below, originally obtained in *Altieri et al., 2021*, have offered the starting point of this work, allowing for a thorough comparison with the same order parameters measured from metagenomic data. Their analytical expressions are self-consistently determined by the system of equations:

$$
\begin{aligned}
h &= \int \mathcal{D}z \left( \frac{\int_0^\infty e^{-\beta \mathcal{H}_{\text{RS}}(N; q_d, q_0, h, z)} N}{\int_0^\infty dN e^{-\beta \mathcal{H}_{\text{RS}}(N; q_d, q_0, h, z)}} \right) = \overline{\langle N \rangle}, \\
q_d &= \int \mathcal{D}z \left( \frac{\int_0^\infty dN e^{-\beta \mathcal{H}_{\text{RS}}(N; q_d, q_0, h, z)} N^2}{\int_0^\infty dN e^{-\beta \mathcal{H}_{\text{RS}}(N; q_d, q_0, h, z)}} \right) = \overline{\langle N^2 \rangle}, \\
q_0 &= \int \mathcal{D}z \left( \frac{\int_0^\infty dN e^{-\beta \mathcal{H}_{\text{RS}}(N; q_d, q_0, h, z)} N}{\int_0^\infty dN e^{-\beta \mathcal{H}_{\text{RS}}(N; q_d, q_0, h, z)}} \right)^2 = \overline{\langle N \rangle^2},
\end{aligned}
\tag{6}
$$

where the calligraphic notation stands for the Gaussian integration $\mathcal{D}z \equiv \int \frac{dz}{\sqrt{2\pi}} e^{-z^2/2}$. In other words, the external average is equivalent to averaging over the quenched disorder $z$, whereas the most

internal one – over the continuous variable $N$ – is interpreted as a thermal average over the single-equilibrium Hamiltonian $\mathcal{H}_{\mathrm{RS}}(N; q_d, q_0, h)$. The latter is denoted by $\langle \cdot \rangle$. See Appendix (Section S2) for more details.

As long as the system of *Equation 6* admits physically reasonable solutions, we might claim that the RS ansatz safely holds. This condition is nevertheless necessary but not sufficient because the stability of the RS solution must also be checked. It therefore requires studying the Hessian matrix of free energy and diagonalizing it on a suitable subspace, called *replicon*, $\mathcal{R}$. The main outcome of this computation is captured by *Equation 3* of the main text. The averaged difference describes the fluctuations between the first and second moments of the species abundances within one state, namely between the diagonal value $q_d$ and the off-diagonal contribution $q_0$ of the overlap matrix. Detecting a vanishing value of $\mathcal{R}$ corresponds to the appearance of a marginally stable RS solution (see Section S3 of the Appendix).

## Moment matching inference

The parameters $h$, $q_0$, and $q_d$ can be self-consistently determined through the saddle point of the dgLV free energy in *Equation 6* (see also *Altieri et al., 2021*). We thus aim to estimate which set of model parameters (i.e., $\boldsymbol{\theta} = (\mu, \sigma, T, \lambda)$) will generate values of the order parameters ($h$, $q_d$, $q_0$) matching those directly estimated from the data. The solution of such an inference problem may not be unique or exact. We have thus developed an optimization algorithm to find a pool of possible solutions that minimize the difference between the order parameters estimated by the model and those directly obtained from the data. To infer the parameters $\boldsymbol{\theta}$, we can thus use the self-consistent equations for the order parameters and solve the inverse problem to find the dgLV parameters that match the empirical observations. For each self-consistent equation, we can introduce a relative error $\delta H(\boldsymbol{\theta}|\boldsymbol{\pi}) = (H(\boldsymbol{\theta}|\boldsymbol{\pi}) - h)/h$, $\delta Q_d(\boldsymbol{\theta}|\boldsymbol{\pi}) = (Q_d(\boldsymbol{\theta}|\boldsymbol{\pi}) - q_d)/q_d$ and $\delta Q_0(\boldsymbol{\theta}|\boldsymbol{\pi}) = (Q_0(\boldsymbol{\theta}|\boldsymbol{\pi}) - q_0)/q_0$. By summing the square of each of these contributions, we introduce the cost function $\mathcal{C}$ for our moment matching problem, that is,

$$\mathcal{C}(\boldsymbol{\theta}|\boldsymbol{\pi}) = \frac{1}{2}\delta H(\boldsymbol{\theta}|\boldsymbol{\pi})^2 + \frac{1}{2}\delta Q_d(\boldsymbol{\theta}|\boldsymbol{\pi})^2 + \frac{1}{2}\delta Q_0^2(\boldsymbol{\theta}|\boldsymbol{\pi}). \tag{7}$$

As already observed, the cost function has multiple local minima. To explore the rich structure of minima, we adopt a greedy search optimization strategy. First, we generate a vector $\boldsymbol{\theta}_0$ so that $m_0 = m(\boldsymbol{\theta}_0|\boldsymbol{\pi}) = 1/2 \max(m) = 1/2$. This condition ensures that the starting point of the optimization is far from the unbounded growth phase. In particular, it allows us to randomly generate an initial value of the interactions heterogeneity from a broad range $\sigma \sim \mathrm{Uniform}(0, 10)$ and to get the corresponding initial value of the demographic noise by means of the relation $T_0 = \frac{(q_d - q_0)\sigma_0^2}{1 - m_0} = 2(q_d - q_0)\sigma_0^2$. The other two parameters are randomly drawn, respectively, as $\mu \sim \mathrm{Uniform}(-1, 1)$ and $\log_{10}\lambda_0 \sim \mathrm{Uniform}(-8, -3)$. The choice of the $\mu_0$ range is justified by the fact that we do not want to bias the interactions to be mutualistic or competitive. Since the unbounded growth phase emerges at $\mu_0 = -1$ (*Biroli et al., 2018*; *Bunin, 2017*), one reasonable choice for the initial condition upper bound is $\mu_0 = 1$. Second, $\lambda_0$ is introduced as a regularizing term for the Langevin dynamics and can be considered small: in this way, we can bound its values as described above. Once the initialization is set, we optimize the cost function. To explore the largest set of solutions, we employ the Broyden–Fletcher–Goldfarb–Shanno algorithm provided by the scipy (*Virtanen et al., 2020*) routine. Briefly, this method allows us to optimize scalar functions of multiple variables using a generalized secant method. Since $\mathcal{C}$ is flat almost everywhere except in the region where local minima are clustered, other methods tend to provide – with the same initialization procedure – results on the boundary of the optimization region, signaling poor convergence performance when tested for our problem. To explore a large subset of solutions and take the flatness of the cost function into account, we repeat the process $10^5$ times and bound the solutions into a region way larger than the initial conditions $\mu \in [-1, 100]$, $\sigma \in [0, 10]$, $T \in [10^{-4}, 10^{-2}]$, and $\lambda \in [10^{-9}, 10^{-1}]$. In the downstream analysis, we only retain the best 30 solutions, minimizing $\mathcal{C}$. At the end of the procedure, we obtain a set of parameters $\boldsymbol{\theta} = (\mu, \sigma, T, \lambda)$ that, if used in the self-consistent equations, are capable of satisfying them with mean relative error $\mathcal{E} = \delta H + \delta Q_d + \delta Q_0$ of order $\mathcal{E} \approx 10^{-2}, 10^{-3}$. As a consistency check, we report the value of $\lambda^* = 2 \times 10^{-6} \approx \min N_{data} = 9 \times 10^{-6}$ (constant for all of the top 30 solutions), which is slightly below the minimum species relative abundance of the data.

## Cavity method for the SAD

Another powerful technique rooted in disordered systems is the cavity method, which turns out to be particularly convenient for deriving the SAD at equilibrium. Without demographic noise, the SAD in the single equilibrium phase is typically captured by a truncated Gaussian distribution (*Yoshino et al., 2008*; *Bunin, 2017*; *Altieri and Franz, 2019*). In the presence of noise and finite migration, the computation gets more involved but is still doable within the cavity approach (*Mezard and Montanari, 2009*).

The basic idea consists of adding a new species to the ecosystem and investigating the resulting joint probability distribution of the typical species. In the thermodynamic limit, the difference between a system composed of $S$ species and the corresponding one with $S + 1$ species is negligible. Therefore, one can write:

$$P_{S+1}(\{N_i\}, N_c) \propto P_S(\{N_i\}) \frac{1}{N_c^{1-\lambda/T}} \exp\left[\frac{1}{T} N_c \left(K - \frac{N_c}{2} - \sum_{j\neq i} \alpha_{cj} N_j\right)\right]. \tag{8}$$

By gathering all relevant information about the so-called *cavity field*, $h_c = \sum_j \alpha_{cj} N_j$, and the higher-order correlation term, we obtain the field distribution, which is defined by the two moments

$$\overline{\overline{h}} = \sum_j \overline{\alpha_{cj}} \overline{\langle N_j \rangle} = \mu h \quad , \quad \overline{\overline{h^2}} = \sum_{j,k} \overline{\alpha_{ci}\alpha_{cj}} \overline{\langle N_i \rangle \langle N_j \rangle} = \sigma^2 q_0. \tag{9}$$

For compactness, we skip all technical details at this stage. Proceeding step-by-step – the full derivation can nevertheless be found in the Appendix – we end up writing the expression for the marginal probability distribution:

$$P_{S+1}(N) \simeq P_S(N) = \int \mathcal{D}\zeta \frac{1}{\mathcal{Z}(\zeta)} N^{\beta\lambda-1} \exp\left\{-\frac{\beta}{2}\left[mN^2 - 2\zeta N\right]\right\}, \tag{10}$$

where $N_c$ has been replaced by $N$ denoting the typical species abundance, $m = \left[1 - \sigma^2\beta(q_d - q_0)\right]$ denotes the *mass* term borrowing field-theory terminology, and $\zeta$ is an auxiliary Gaussian variable.

## Acknowledgements

We thank Silvia De Monte for insightful discussions. SS acknowledges Iniziativa PNC0000002-DARE – Digital Lifelong Prevention. AM acknowledges financial support under the National Recovery and Resilience Plan (NRRP), Mission 4, Component 2, Investment 1.1, Call for tender No. 104 by the Italian Ministry of University and Research (MUR), funded by the European Union – NextGenerationEU – Project Title 'Emergent Dynamical Patterns of Disordered Systems with Applications to Natural Communities' – CUP 2022WPHMXK – Grant Assignment Decree No. 2022WPHMXK adopted on 19/09/2023 by the Italian Ministry of Ministry of University and Research (MUR). AA acknowledges the support received from the Agence Nationale de la Recherche (ANR), under the grant ANR-23-CE30-0012-01 (project 'SIDECAR').

## Additional information

### Funding

| Funder | Grant reference number | Author |
| --- | --- | --- |
| Agence Nationale de la Recherche | ANR-23-CE30-0012-01 | Ada Altieri |
| National Recovery and Resilience Plan | CUP 2022WPHMXK | Samir Suweis |
| DigitalLifelong Prevention | PNC0000002-DARE | Samir Suweis |

| Funder | Grant reference number | Author |
|---|---|---|

The funders had no role in study design, data collection and interpretation, or the decision to submit the work for publication.

## Author contributions

Jacopo Pasqualini, Formal analysis, Methodology, Writing – original draft; Amos Maritan, Formal analysis, Visualization, Writing – original draft; Andrea Rinaldo, Resources, Visualization; Sonia Facchin, Edoardo Vincenzo Savarino, Data curation, Visualization; Ada Altieri, Conceptualization, Formal analysis, Supervision, Writing – original draft, Writing – review and editing, Methodology; Samir Suweis, Conceptualization, Supervision, Writing – original draft, Writing – review and editing

## Author ORCIDs

Jacopo Pasqualini ⓘ https://orcid.org/0009-0005-2579-7321
Amos Maritan ⓘ https://orcid.org/0000-0002-3535-7873
Sonia Facchin ⓘ https://orcid.org/0000-0002-6774-590X
Ada Altieri ⓘ https://orcid.org/0000-0002-7750-2178
Samir Suweis ⓘ https://orcid.org/0000-0002-1603-8375

Reviewer #1 (Public review): https://doi.org/10.7554/eLife.105948.3.sa1
Reviewer #2 (Public review): https://doi.org/10.7554/eLife.105948.3.sa2
Reviewer #3 (Public review): https://doi.org/10.7554/eLife.105948.3.sa3
Author response https://doi.org/10.7554/eLife.105948.3.sa4

# Additional files

## Supplementary files
MDAR checklist

## Data availability
Data and parameters obtained from the moment matching inference are available at Zenodo: https://doi.org/10.5281/zenodo.11934376.

The following dataset was generated:

| Author(s) | Year | Dataset title | Dataset URL | Database and Identifier |
|---|---|---|---|---|
| Pasqualini J | 2024 | Microbiomes Through the Looking Glass | https://doi.org/10.5281/zenodo.11934376 | Zenodo, 10.5281/zenodo.11934376 |

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

## Appendix 1

### S1. Mapping between MacArthur and Lotka–Volterra models

We aim to investigate the mapping between the Lotka–Volterra model and the consumer–resource (CR) or MacArthur (MA) dynamics, as analyzed in *Tikhonov and Monasson, 2017*; *Altieri and Franz, 2019* for a high-dimensional version. We thus consider *Equation 1* of the main text, which describes the evolution of $S$ randomly interacting species. The self-interaction term parameters are the growth rate $r_i$, and the carrying capacity $K_i$, whereas all the inter-species interaction parameters are collected into the matrix $\boldsymbol{\alpha}$. For this argument, we will consider a simplified model version, where stochasticity (a.k.a. effective temperature) and immigration rate are set to zero, leading to a deterministic version, as in *Biroli et al., 2018*. This model describes the dynamics of the relative species abundances $N_i$ – previously normalized with respect to the total number of individuals in the pool – according to:

$$\frac{\dot{N}_i}{N_i} = \frac{r_i}{K_i}(K_i - N_i) - \sum_{j \neq i} \alpha_{i,j} N_j = r_i - \frac{r_i}{K_i} N_i - \sum_{j \neq i} \alpha_{i,j} N_j = \rho K_i - \rho N_i - \sum_{j \neq i} \alpha_{i,j} N_j \,, \tag{11}$$

where we have introduced $\frac{r_i}{K_i} = \rho_i = \rho$, the proportionality constant between growth rate and carrying capacity. Our goal is to re-derive here a generalized Lotka–Volterra (gLV) model with symmetric interactions and therefore to justify the assumption adopted in the main text. To do that, we introduce the MA model, which describes how $S$ species compete to grab $M$ available resources. For more details, see also *Wu et al., 2021*. We thus consider the following set of coupled ODEs, describing the joint dynamics of resources abundances $R_\alpha$ and species abundances $N_i$:

$$\begin{cases} \dot{N}_i = N_i \left( \sum_\beta C_{i,\beta} R_\beta - m_i \right) \\ \dot{R}_\alpha = f_\alpha(R_\alpha) - R_\alpha \sum_j C^T_{\alpha,j} N_j \,, \end{cases} \tag{12}$$

where $f_\alpha = R_\alpha(\tilde{K}_\alpha - R_\alpha)$ describes resources logistic regulation in the absence of consumer species. The parameters here are the death rate of each species $m_i$, the carrying capacity of each resource $\tilde{K}_\alpha$, and the uptake matrix $\boldsymbol{C}$, which describes the consumption or production versus microbial growth inhibition of the resource $\beta$ per unit of time by the species $i$. Each row of the matrix can be considered an $M$-dimensional vector, $\vec{C}_i$, which we will refer to as the uptake profile of the species $i$. The key point is that such vectors might have negative entries, as it happens in the case of microbial growth inhibition. If the dynamic of resources is much faster than the one of species, we can consider the associated equation in the stationary state. With this approximation, provided that the resources abundance is larger than zero ($R_\alpha > 0$), we can evaluate the stationary value of the resource abundances

$$\tilde{K}_\alpha - R_\alpha - \sum_j C_{j,\alpha} N_j \longrightarrow R_\alpha = \tilde{K}_\alpha - \sum_j C_{j,\alpha} N_j \,. \tag{13}$$

In microbial systems, resources — such as sugars, vitamins, and other metabolites — continuously flow into the system, are consumed by the community, and are diluted, but they do not necessarily follow logistic growth. For microbes, a more meaningful justification of the gLV model from a CR perspective arises from considering CR dynamics in a chemostat, where a separation of timescales is assumed and higher-order interactions are neglected (*Posfai et al., 2017*; *Goyal et al., 2025*).

By inserting the resource abundance resulting from the logistic approximation in the equation for species dynamics, we precisely recover the *gLV* model:

$$\begin{aligned} \frac{\dot{N}_i}{N_i} &= \sum_\beta \left[ C_{i,\beta} \left( \tilde{K}_\beta - \sum_j C_{j,\beta} N_j \right) \right] - m_i = \left( \sum_\beta C_{i,\beta} \tilde{K}_\beta - m_i \right) - \sum_\beta \left( \sum_j C_{i,\beta} C^T_{\beta,j} \right) N_j \\ &= r_i^{MA} - \left( \sum_\beta C_{i,\beta} C^T_{\beta,i} \right) N_i - \sum_{j \neq i} \left( \sum_\beta C_{i,\beta} C^T_{\beta,j} \right) N_j = r_i^{MA} - A_{i,i} N_i - \sum_{j \neq i} A_{i,j} N_j \,. \end{aligned} \tag{14}$$

Note that, between the second and third passage, we decomposed the sum over $j$ in the diagonal and off-diagonal terms, respectively, $j = i$ and $j \neq i$, then swapped the summation over resources and species. Finally, we have introduced the quantities

$$r_i^{MA} = \sum_\beta C_{i,\beta} \tilde{K}_\beta - m_i, \quad A_{i,j} = \sum_\beta C_{i,\beta} C_{\beta,j}^T . \tag{15}$$

Since the interactions are expressed as a scalar product between uptake profiles, the derived equations correspond to a gLV model with symmetric interactions.

## S2. Replica formalism for the dgLV model

To obtain a full characterization of the different emergent phases in the dgLV model, we can take advantage of the replica method. Using the replica identity (*Mézard et al., 1987*; *Altieri and Baity-Jesi, 2024*), that is $\overline{\ln Z} = \lim_{n \to 0} \frac{\overline{Z^n} - 1}{n}$ – which has been rigorously proven in some specific instances (*Guerra and Toninelli, 2002*; *Panchenko and Talagrand, 2007*; *Dia et al., 2016*) – the disordered average is directly computed on the replicated partition function, for $n$ replicas of the same sample. The index $n$ is initially treated as an integer; however, relying on the assumption that the analytical continuation $n \to 0$ exists.

$$\overline{Z^n} = \int \prod_{i<j} d\alpha_{ij} \exp\left[ -\sum_{i<j} \frac{(\alpha_{ij} - \mu/S)^2}{2\sigma^2/S} \right] \int \prod_{a=1}^{n} \prod_i dN_i^a \exp\left[ -\beta H(\{N_i^a\}) \right] , \tag{16}$$

where $\beta = 1/T$ is the inverse of the demographic noise amplitude. The overline $\overline{\cdot}$ in the partition function denotes the average over the disorder, namely over the Gaussian variables $\alpha_{ij}$ with finite mean $\mu/S$ and variance $\sigma^2/S$. The index $a$ takes the number of replicas into account, with $a = 1, ..., n$.

The analytical treatment requires the introduction of the overlap matrix, $Q_{ab}$ – whose diagonal value is $Q_{aa} - Q_{ab} = \frac{1}{S} \sum_{i=1}^{S} N_i^a N_i^b$, which can be easily parametrized in the replica space, as well as the average abundance $h_a = \frac{1}{S} \sum_{i=1}^{S} N_i^a$. For the latter, we will consider the uniformity condition, with $h_a = h$, $\forall a$. We can thus re-express the free energy as a function of the aforementioned order parameters

$$F = -\frac{1}{\beta n} \ln \int \prod_{a,(a<b)} dQ_{ab} dQ_{aa} dh_a \, e^{S \mathcal{A}(Q_{ab}, Q_{aa}, h_a)}, \tag{17}$$

which remains, at this level, as general as possible. The action $\mathcal{A}$ reads

$$\mathcal{A}(Q_{ab}, Q_{aa}, h_a) = -\rho^2 \sigma^2 \beta^2 \sum_{a<b} \frac{Q_{ab}^2}{2} - \rho^2 \sigma^2 \beta^2 \sum_a \frac{Q_{aa}^2}{4} + \rho \mu \beta \sum_a \frac{h_a^2}{2} + \frac{1}{S} \sum_i \ln Z_i, \tag{18}$$

to be eventually evaluated by the Laplace method, or saddle-point approximation, in the large $S$ limit.

By resorting to the replica-symmetric (RS) ansatz – according to which the overlap matrix is parametrized by two values only, $q_d$ and $q_0$, the diagonal value or self-overlap, and the off-diagonal value, respectively – the last piece in *Equation 18* can be expressed as a function of an effective Hamiltonian $\mathcal{H}_{\text{eff}}(\{N_i^a\})$. It essentially embeds the contribution of the quadratic Lotka–Volterra potential and the additional Lagrange multipliers enforcing the above expressions for $Q_{ab}, Q_{aa}$, and $h_a$.

$$\mathcal{H}_{\text{eff}}(\{N_i^a\}) = -\frac{\rho^2 \sigma^2 \beta}{2}(q_d - q_0) \sum_a (N_i^a)^2 - \frac{\rho^2 \sigma^2 \beta}{2} q_0 \left( \sum_a N_i^a \right)^2 + \sum_a \rho \mu h N_i^a + \sum_a V_i(N_i^a) + (T - \lambda) \sum_a \ln(N_i^a),$$

$$\tag{19}$$

The free energy is thus re-written as $F = -\lim_{n \to 0} \ln Z^n/(\beta n) = -\lim_{n \to 0} \mathcal{A}(q_d, q_0, h)/(\beta n)$. However, integrating the quenched disorder out implies a further complication: replica indices turn out to be coupled, as is evident in the second term above. The next step will then require the introduction of an auxiliary Gaussian variable $z$, with zero mean and unit variance so that:

$$Z_i = \int_{-\infty}^{+\infty} \frac{dz_i}{\sqrt{2\pi}} e^{-z_i^2/2} \int \prod_{a=1}^{n} dN_i^a e^{-\beta \sum_a \mathcal{H}_{\mathrm{RS}}(N_i^a; q_d, q_0, h, z_i)}, \tag{20}$$

with the associated RS Hamiltonian:

$$\begin{aligned}\mathcal{H}_{\mathrm{RS}}\left(N_i^a, z_i\right) &= -\rho^2 \sigma^2 \beta \left(q_d - q_0\right) \frac{\left(N_i^a\right)^2}{2} + \left(\rho\mu h - z_i \rho \sqrt{q_0} \sigma\right) N_i^a + V_i\left(N_i^a\right) + (T - \lambda) \ln N_i^a \\ &= \frac{\left(N_i^a\right)^2}{2} \left[\rho - \rho^2 \sigma^2 \beta \left(q_d - q_0\right)\right] + \left(\rho\mu h - z_i \rho \sigma \sqrt{q_0} - \rho K_i\right) N_i^a + (T - \lambda) \ln N_i^a.\end{aligned} \tag{21}$$

Note that, although in the computation above we have considered a species-dependent $K_i$ to be as general as possible, in our work we have assumed $K_i = K$ for all species. The full expressions of the parameters $h$, $q_0$, and $q_d$, to be obtained by a saddle-point approximation of the RS free energy have been reported in the Methods. They can be solved iteratively, as also explained in *Altieri et al., 2021*; *Lorenzana and Altieri, 2022*.

### Comment on compositional abundances

For purely theoretical purposes, dealing with absolute or relative abundances makes little difference. However, for the metagenomics data available to us, it is essential to perform the analysis as a function of relative abundances. Forcing the abundances to be normalized is formally equivalent to adding a global constraint in the Hamiltonian such that $\sum_i N_i = S$ through the Lagrange multiplier $\gamma$.

Accordingly, the original partition function can be recast as

$$\bar{Z} = \int \prod_{i<j} d\alpha_{ij} \exp\left[-\sum_{i<j} \frac{(\alpha_{ij} - \mu/S)^2}{2\sigma^2/S}\right] \int \prod_i dN_i \int_{-i\infty}^{i\infty} d\gamma \, \exp\left[-\beta H(\{N_i\}) - \gamma \sum_i (N_i - 1)\right]. \tag{22}$$

The procedure explained above remains exactly the same along with the introduction of $n$ replicas of the reference system. Optimizing over $Q_{ab}$, $H_a$, and the parameter $\gamma_a$ things get easier within the RS approximation. In fact, $\gamma_a = \gamma$, satisfying a uniformity condition for all replicas. The only visible difference is in the linear term of (*Equation 21*) which now incorporates the explicit dependence on the multiplier $\gamma$ leading to

$$\mathcal{H}_{\mathrm{RS}}(N_i^a, z_i) = \frac{(N_i^a)^2}{2} \left[\rho - \rho^2 \sigma^2 \beta(q_d - q_0)\right] + \left(\rho\mu h - z_i \rho \sigma \sqrt{q_0} - \rho K_i - \gamma\right) N_i^a + (T - \lambda) \ln N_i^a. \tag{23}$$

We have already discussed in the text how the $\mu$ parameter does not play a significant role in determining the different phases, confirming prior results by *Bunin, 2017*; *Lorenzana and Altieri, 2022*. Once more, we can safely claim that the actual differences and emerging data clustering between healthy and sick samples are driven by the noise amplitude $T$ and the interaction heterogeneity $\sigma$.

## S3. Stability analysis: Hessian matrix and zero modes

To properly understand the stability of the single equilibrium phase and highlight possibly emergent multiple attractor regimes, we need to perform a stability computation. Computing the stability against external perturbations requires first the definition of the replicated partition function:

$$\begin{aligned}Z_i &= \int \prod_a dN_i^a \exp\left[\frac{\beta^2 \rho^2 \sigma^2}{2} \sum_{a<b} Q_{ab} N_i^a N_i^b + \beta^2 \rho^2 \sigma^2 \sum_a (N_i^a)^2 \frac{Q_{aa}}{2} - \rho\beta\mu \sum_a N_i^a h^a - \beta \sum_a V_i(N_i^a) \right. \\ &\quad \left. \pm \beta \sum_a (T - \lambda) \ln N_i^a\right],\end{aligned} \tag{24}$$

then the study of the harmonic fluctuation of the free energy around the RS solution. For the latter, the Hessian matrix of the action $\mathcal{A}$ is needed

$$\mathcal{M}_{abcd} \equiv -\frac{\partial^2 \mathcal{A}}{\partial Q_{ab} \partial Q_{cd}} = \beta^2 \rho^2 \sigma^2 \left[\delta_{(ab),(cd)} - (\beta^2 \rho^2 \sigma^2)\overline{\langle N^a N^b, N^c N^d \rangle_c}\right], \tag{25}$$

where the subscript $\langle \cdot \rangle_c$ denotes the connected part of the correlation function. Leveraging well-known techniques in field theory and disordered systems (*Dominicis and Giardina, 2006*), we decompose the Hessian matrix (*Equation 25*) as a function of three different correlators among which the one living in the replicon subspace (*Dominicis and Giardina, 2006*; *Altieri et al., 2016*) provides the leading contribution:

$$\mathcal{R} = (\beta\rho\sigma)^2 \left[ 1 - (\beta\rho\sigma)^2 \left( \tilde{M}_{ab,ab} - 2\tilde{M}_{ab,ac} + \tilde{M}_{ab,cd} \right) \right]. \tag{26}$$

Three different elements appear in the computation depending on the constraint/equality condition of their replica indices

$$\tilde{M}_{ab,ab} - 2\tilde{M}_{ab,ac} + \tilde{M}_{ab,cd} = \left[ \overline{\langle (N^a)^2 (N^b)^2 \rangle} - 2\overline{\langle (N^a)^2 N^b N^c \rangle} + \overline{\langle N^a N^b N^c N^d \rangle} \right]. \tag{27}$$

Sticking to a single-equilibrium regime, namely in the RS approximation, the expression for the replicon eigenvalue can be further simplified

$$R = (\beta\rho\sigma)^2 \left[ 1 - (\beta\rho\sigma)^2 \overline{\left( \langle N^2 \rangle - \langle N \rangle^2 \right)^2} \right], \tag{28}$$

which precisely corresponds to *Equation 3* of the main text. A strictly positive or a vanishing value of the replicon is associated with a stable or marginally stable phase, respectively. In the second case, the RS ansatz is no longer valid and requires the use of a one- or multiple-step replica-symmetry breaking.

## S4. Cavity argument for the species abundance distribution

In the absence of demographic fluctuations and immigration, it is well-known that the species abundance distribution (SAD) reflects a truncated Gaussian (*Yoshino et al., 2008*; *Bunin, 2017*; *Tikhonov and Monasson, 2017*; *Lorenzana and Altieri, 2022*). Albeit the computation gets more complicated in the presence of demographic noise, we can nevertheless employ the cavity method (*Mezard and Montanari, 2009*; *Zamponi, 2010*; *Altieri and Baity-Jesi, 2024*) to obtain the single-species marginal probability distribution. Based on a cavity argument, we pretend to add a new species to the ecosystem and eventually investigate the resulting joint probability distribution. In the thermodynamic limit, for $S \gg 1$, it reads

$$P_{S+1}(\{N_i\}, N_c) \propto P_S(\{N_i\}) \frac{1}{N_c^{1-\lambda/T}} \exp\left[ \frac{1}{T} N_c \left( K - \frac{N_c}{2} - \sum_{j\neq i} \alpha_{cj} N_j \right) \right], \tag{29}$$

where, for the sake of simplicity, we have assumed the carrying capacities to be species-independent, that is $K_i = K$.

The last piece in parentheses denotes the *cavity field*, $h_c = \sum_j \alpha_{cj} N_j$ which, as long as the single equilibrium phase is concerned, can be assumed to be a Gaussian random variable. The probability distribution is formally factorized and the only coupling with the new species occurs via $h_c$:

$$P_{S+1}(N_c) \propto \int_0^\infty \prod_i dN_i P_S(\{N_i\}) \frac{1}{N_c^{1-\lambda/T}} \exp\left[ \frac{1}{T} N_c \left( K - \frac{N_c}{2} - \sum_{j\neq i} \alpha_{cj} N_j \right) \right]. \tag{30}$$

The probability distribution conditioned to $h_c$ then reads:

$$P_{S+1}(N_c|h_c) \propto \frac{1}{N_c^{1-\lambda/T}} \exp\left[ \frac{1}{T} N_c \left( K - \frac{N_c}{2} - h_c \right) \right] P(h_c), \tag{31}$$

where the thermal and the disordered averages of the cavity field, $h_c$, must eventually be evaluated. Furthermore, to go beyond a naïve mean-field approximation, we need to subtract the background effect exerted by all other species on the picked one through an *Onsager reaction term*.

$$P_{S+1}(N_c) = \frac{1}{Z} \int_{-\infty}^{+\infty} \frac{dh_c}{\sqrt{2\pi \, \text{var}[h_c]}} \frac{1}{N_c^{1-\lambda/T}} \exp\left[\frac{1}{T} N_c \left(K - \frac{N_c}{2} - h_c\right)\right] \exp\left(-\frac{(h_c - \tilde{h})^2}{2 \, \text{var}[h_c]}\right)$$

$$= \frac{1}{Z} \frac{1}{N_c^{1-\lambda/T}} \exp\left\{-\frac{N_c}{2T^2} \left[N_c(T - \text{var}[h_c]) - 2KT + 2\tilde{h}T\right]\right\},$$

(32)

In the above expression, $\tilde{h} = \langle h_c \rangle = \sum_j \alpha_{cj} \langle N_j \rangle$ and $\text{var}[h_c] = \langle h_c^2 \rangle - \langle h_c \rangle^2$ stand, respectively, for the first moment and the variance of the field with respect to the thermal average, in the thermodynamic limit. The normalization, $Z$, can formally be expressed as a combination of error and hypergeometric functions as follows:

$$Z = \int_0^\infty \frac{dN_c}{N_c^{1-\lambda/T}} \exp\left\{-\frac{N_c}{2T^2} \left[N_c(T - \text{var}[h_c]) - 2KT + 2\tilde{h}T\right]\right\} =$$

$$= 2^{-1+\frac{\lambda}{2T}} \left(\frac{T - \text{var}[h_c]}{T^2}\right)^{-\frac{\lambda}{2T}} \left(\text{Gamma}\left[\frac{\lambda}{2T}\right] \text{HypergeometricF1}\left[\frac{\lambda}{2T}, \frac{1}{2}, \frac{(K-\tilde{h})^2}{2(T - \text{var}[h_c])}\right] + \right.$$

$$\left. \frac{\sqrt{2}(-\tilde{h} + K)}{\sqrt{T - \text{var}[h_c]}} \text{Gamma}\left[\frac{T+\lambda}{2T}\right] \text{HypergeometricF1}\left[\frac{T+\lambda}{2T}, \frac{3}{2}, \frac{(K-\tilde{h})^2}{2(T - \text{var}[h_c])}\right]\right).$$

(33)

The next step requires the computation of the disordered average. By doing so, we end up with the distribution of the second field, which is defined in terms of the two moments:

$$\overline{\tilde{h}} = \sum_j \overline{\alpha_{cj}} \langle N_j \rangle = \mu h, \qquad \overline{\tilde{h}^2} = \sum_{j,k} \overline{\alpha_{ci}\alpha_{cj}} \langle N_i \rangle \langle N_j \rangle = \sigma^2 q_0.$$

(34)

A similar derivation was obtained in the case of a gLV model with noise to be compared with numerical distributions in different regimes (**Wu et al., 2021**). The resulting SAD typically exhibits a divergence at $N = 0$ and a secondary maximum, therefore closer to what is referred to as *niche scenario*.

Then, by introducing a rescaled random variable $\zeta$ and integrating over it, the marginal probability distribution becomes

$$P_{S+1}(N_c) = \int_{-\infty}^{\infty} \frac{d\zeta}{\sigma\sqrt{2\pi q_0}} e^{-\frac{[\zeta - (K - \mu h)]^2}{2\sigma^2 q_0}} \frac{1}{\mathcal{Z}(\zeta) N_c^{1-\lambda/T}} \exp\left\{-\frac{\beta}{2}\left[N_c^2\left(1 - \sigma^2\beta(q_d - q_0)\right) - 2N_c\zeta\right]\right\}$$

$$\to \int \mathcal{D}\zeta \frac{1}{\mathcal{Z}(\zeta)} N^{\beta\lambda - 1} \exp\left\{-\frac{\beta}{2}\left[mN^2 - 2\zeta N\right]\right\},$$

(35)

where $N_c$ is just replaced by $N$ denoting the typical species abundance and $T \to 1/\beta$. The expression for the normalization is roughly similar to the one in **Equation 32** with the only difference being that the mean and the variance of the cavity field must be replaced by the quenched average of $\tilde{h}$ and $\sigma^2(q_d - q_0)$, respectively.

## Appendix 2

### Estimation of neutral and niche processes

Following the results of the cavity method in Section D, we can describe the SAD of the system (see *Equation 35*) via the quantities $\nu = \beta\lambda > 0$, the quadratic coefficient $m = 1 - \sigma^2\beta(q_d - q_0)$ and the disorder variable $\zeta = K - \mu h + \sqrt{q_0}\sigma z$, where $z$ is a standardized normal variable.

As we specified in the main text, we refer to the quadratic coefficient $m$ as a mass. When the mass becomes negative, the self-consistent *Equation 6* in the Methods section is no longer well-defined, and the order parameters diverge. This signals a transition to an unbounded growth regime.

So far, the obtained SAD is flexible enough to accommodate different qualitative shapes frequently appearing in theoretical ecology (*Azaele et al., 2016*). When $\nu > 1$, the migration strength turns out to be stronger than demographic fluctuations and the distribution resembles a modal log-normal. On the other hand, when $0 < \nu < 1$ demographic noise overcomes immigration, a low-abundance (integrable) divergence appears. In our case, $\nu \approx 10^{-3}$, so demographic noise is much stronger than immigration.

When the divergence in zero appears, two further shapes are possible. If there exists a secondary extremum for the SAD, then there is a coexistence of niche (driven by species interactions) and neutral (driven by demographic noise) effects. Conversely, if no secondary extrema exist, birth and death effects are the key drivers of the dynamics. In our interpretation of the model, the SAD is the stationary distribution of a *single* realization of the Lotka–Volterra dynamics. Given a set of parameters $\boldsymbol{\theta} = (\mu, \sigma, \beta, \lambda)$, it is possible to evaluate how many realizations of the Lotka–Volterra dynamics will display a secondary extremum. Taking the form of a fraction, we introduce a quantity $\psi$ that captures the chance of displaying such an extremum and measures the relative contribution between selective and neutral effects. Namely, we can evaluate the probability that $\mathcal{H}_{\mathrm{RS}}$ has a minimum different from zero:

$$\frac{\partial p(N|\zeta)}{\partial N} = -\beta\frac{\partial\mathcal{H}_{\mathrm{RS}}}{\partial N}p(N|\zeta) = 0 \rightarrow \frac{\partial\mathcal{H}_{\mathrm{RS}}}{\partial N} = \frac{1-\nu}{\beta N} + mN - \zeta = 0 \,. \tag{36}$$

Being an algebraic equation of degree two, it has two solutions $N^*_\pm = \frac{\zeta \pm \sqrt{\zeta^2 - 4m(1-\nu)/\beta}}{2m}$. Introducing $\zeta^* = \sqrt{\frac{4(1-\nu)m}{\beta}}$, the condition for the local extrema to appear if $\zeta^2 > \frac{4m(1-\nu)}{\beta} = (\zeta^*)^2 > 0$. This probability amounts to

$$\psi = \mathbb{P}[\zeta^2 > (\zeta^*)^2] = \mathbb{P}\left[\zeta > \zeta^*\right] + \mathbb{P}\left[\zeta < -\zeta^*\right] = \frac{1}{2}\mathrm{Erfc}\left(\frac{\zeta^* + \overline{\zeta}}{\sqrt{2}\sigma_\zeta}\right) + \frac{1}{2}\mathrm{Erfc}\left(\frac{\zeta^* - \overline{\zeta}}{\sqrt{2}\sigma_\zeta}\right) \,, \tag{37}$$

where $\overline{\zeta} = K - \mu h$, $\sigma_\zeta = \sqrt{q_0}\sigma$ are the moments of the variable tracking the different disorder realizations. When the quantity $\psi$ is of order 1, niche and neutral contributions to the dynamics can be considered comparable. In principle, we should check which of the two ± solutions is actually a maximum. The convexity of the SAD is easy to evaluate:

$$\begin{aligned}\frac{\partial^2 p(N \mid \zeta)}{\partial N^2} &= -\beta\left(\left(\frac{\partial^2 H_{\mathrm{RS}}}{\partial N^2} - \beta\left(\frac{\partial H_{\mathrm{RS}}}{\partial N}\right)^2\right)p(N \mid \zeta)\right)\Big|_{N=N^*_\pm} \\ &= \left(\frac{\partial^2 H_{\mathrm{RS}}}{\partial N^2} - \beta\left(\frac{\partial H_{\mathrm{RS}}}{\partial N}\right)\right)\Big|_{N^*_\pm} = 2m\left(1 + \frac{1}{y}\left(1 \mp \sqrt{1-y}\right)\right) < 0\end{aligned} \tag{38}$$

where $y = (\frac{\zeta^*}{\zeta})^2$. From the first inequality, straightforward algebraic manipulations confirm that $N^*_+$ is the stable solution. Intuitively, since $P(N|\zeta)$ has a divergence in zero, any secondary maximum $N^*_+$ – if it exists – must be greater than the minimum $N^*_-$.

