## [Editor Report · eLife Assessment]

This **important** study shows how the relative importance of inter-species interactions in microbiomes can be inferred from empirical species abundance data. The methods based on statistical physics of disordered systems are **compelling** and rigorous, and allow for distinguishing healthy and non-healthy human gut microbiomes via differences in their inter-species interaction patterns. This work should be of broad interest to researchers in microbial ecology and theoretical biophysics.

---

## [Referee Report · Reviewer #1 (Public review)]

Summary:

In this manuscript, the authors develop a novel method to infer ecologically-informative parameters across healthy and diseased states of the gut microbiota, although the method is generalizable to other datasets for species abundances. The authors leverage techniques from theoretical physics of disordered systems to infer different parameters-mean and standard deviation for the strength of bacterial interspecies interactions, a bacterial immigration rate, and the strength of demographic noise-that describe the statistics of microbiota samples from two groups-one for healthy subjects and another one for subjects with chronic inflammation syndromes. To do this, the authors simulate communities with a modified version of the Generalized Lotka-Volterra model and randomly-generated interactions, and then use a moment-matching algorithm to find sets of parameters that better reproduce the data for species abundances. They find that these parameters are different for the healthy and diseased microbiota groups. The results suggest, for example, that bacterial interaction strengths, relative to noise and immigration, are more dominant of microbiota dynamics in diseased states than in healthy states.

We think that this manuscript brings an important contribution that will be of interest in the areas of statistical physics, (microbiota) ecology and (biological) data science. The evidence of their results is solid and the work improves the state-of-the-art in terms of methods.

Strengths:

Using a fairly generic ecological model, the method can identify the change in the relative importance of different ecological forces (distribution of interspecies interactions, demographic noise and immigration) in different sample groups. The authors focus on the case of the human gut microbiota, showing that the data is consistent with a higher influence of species interactions (relative to demographic noise and immigration) in a disease microbiota state than in healthy ones.The method is novel, original and it improves the state-of-the-art methodology for the inference of ecologically-relevant parameters. The analysis provides solid evidence on the conclusions.

Weaknesses:

As a proof of concept for a new inference method, this text maintains a technical focus, which may require some familiarity with statistical physics. Nevertheless, the authors' clear introduction of key mathematical terms and their interpretations, along with a clear discussion of the ecological implications, make the results accessible and easy to follow.

---

## [Referee Report · Reviewer #2 (Public review)]

Summary:

This valuable work aims to infer, from microbiome data, microbial species interaction patterns associated with healthy and unhealthy human gut microbiomes. Using solid techniques from statistical physics, the authors propose that healthy and unhealthy microbiome interaction patterns substantially differ. Unhealthy microbiomes are closer to instability and single-strain dominance; whereas healthy microbiomes showcase near-neutral dynamics, mostly driven by demographic noise and immigration.

Strengths:

This is a well-written article, relatively easy to follow and transparent despite the high degree of technicality of the underlying theory. The authors provide a powerful inferring procedure, which bypasses the issue of having only compositional data. This work shows that embracing the complexity of microbial systems can be used to our advantage, instead of being an insurmountable obstacle. This is a powerful counterpoint to the classic reductionist view that pushes researchers to study much simpler systems, and only hope to one day scale up their findings.

Weaknesses:

As acknowledged by the authors themselves, this is only a proof of concept. Further research is to better understand the dynamical nature of gut-microbiomes. The authors do however point at ways in which species abundance distributions could be better reproduced by dynamical models. They also suggest that they work could explain prior empirical findings invoking the "Anna Karenina principle", wherehealthy microbiomes resemble one another, but disease states tend to all differ.

---

## [Referee Report · Reviewer #3 (Public review)]

Summary:

I found the manuscript to be well-written. I have a few questions regarding the model, though the bulk of my comments are requests to provide definitions and additional clarity. There are concepts and approaches used in this manuscript that are clear boons for understanding the ecology of microbiomes but are rarely considered by researchers approaching the manuscript from a traditional biology background. The authors have clearly considered this in their writing of S1 and S2, so addressing these comments should be straightforward. The methods section is particularly informative and well-written, with sufficient explanations of each step of the derivation that should be informative to researchers in the microbial life sciences that are not well-versed with physics-inspired approaches to ecology dynamics.

Strengths:

The modeling efforts of this study primarily rely on a disordered for of the generalized Lotka-Volterra (gLV) model. This model can be appropriate for investigating certain systems and the authors are clear about when and how more mechanistic models (i.e., consumer-resource) can lead to gLV. Phenomenological models such as this have been found to be highly useful for investigating the ecology of microbiomes, so this modeling choice seems justified, and the limitations are laid out.

Weaknesses:

The authors use metagenomic data of diseased and healthy patients that was first processed in Pasqualini et al. (2024). The use of metagenomic data leads me into a question regarding the role of sampling effort (i.e., read counts) in shaping model parameters such as $h$. This parameter is equal to the average of 1/# species across samples because the data are compositional in nature. My understanding is that $h$ was calculated using total abundances (i.e., read counts). The number of observed species is strongly influenced by sampling effort and the authors addressed this point in their revised manuscript.

However, the role of sampling effort can depend on the type of data and my instinct about the role that sampling effort plays in species detection is primarily based on 16S data. The dependency between these two variables may be less severe for the authors' metagenomic pipeline. This potential discrepancy raises a broader issue regarding the investigation of microbial macroecological patterns and the inference of ecological parameters. Often microbial macroecology researchers rely on 16S rRNA amplicon data because that type of data is abundant and comparatively low-cost. Some in microbiology and bioinformatics are increasingly pushing researchers to choose metagenomics over 16S. Sometimes this choice is valid (discovery of new MAGs, investigate allele frequency changes within species, etc.), sometimes it is driven by the false equivalence "more data = better". The outcome though is that we have a body of more-or-less established microbial macroecological patterns which rest on 16S data and are now slowly incorporating results from metagenomics. To my knowledge there has not been a systematic evaluation of the macroecological patterns that do and do not vary by one's choice in 16S vs. metagenomics. Several of the authors in this manuscript have previously compared the MAD shape for 16S and metagenomic datasets in Pasqualini et al., but moving forward a more comprehensive study seems necessary (2024). These points were addressed by the authors in their revised manuscript.

Final review: The authors addressed all comments and I have no additional comments.

References

Pasqualini, Jacopo, et al. "Emergent ecological patterns and modelling of gut microbiomes in health and in disease." PLOS Computational Biology 20.9 (2024): e1012482.

---

## [Author Response]

The following is the authors’ response to the original reviews

**Recommendations for the Authors:**

**Reviewer #1:**

We think that this manuscript brings an important contribution that will be of interest in the areas of statistical physicists, (microbiota) ecology, and (biological) data science. The evidence of their results is solid and the work improves the state-of-the-art in terms of methods. We have a few concerns that, in our opinion, the authors should address.

Major concerns:(1) While the paper could be of interest for the broad audience of e-Life, the way it is written is accessible mainly to physicists. We encourage the authors to take the broad audience into account by (i) explaining better the essence of what is being done at each step, (ii) highlighting the relevance of the method compared to other methods, (iii) discussing the ecological implications of the results.Examples on how to approach (i) include: Modify or expand Figure 1 so that non-familiar readers can understand the summary of the work (e.g. with cartoons representing communities, diseased states and bacterial interactions and their relationship with the inference method); in each section, summarize at the beginning the purpose of what is going to be addressed in this section, and summarize at the end what the section has achieved; in Figure 2, replace symbols by their meaning as much as possible-the same for Figure 1, at the very least in the figure caption.Example on how to approach (ii): Since the authors aim to establish a bridge between disordered systems and microbiome ecology, it could be useful to expand a bit the introduction on disordered systems for biologists/biophysicists. This could be done with an additional text box, which could also highlight the advantages of this approach in comparison to other techniques (e.g. model-free approaches can also classify healthy and diseased states).Example on how to approach (iii): The authors could discuss with more depth the ecological implications of their results. For example, do they have a hypothesis on why demographic and neutral effects could dominate in healthy patients?

We thank the reviewer for the observations. Following the suggestion in the revised version, each section outlines the goal of what will be addressed in that section, and summarizes what we have achieved at the end; We also updated Figure 1 and Figure 2.

(i) For figure 1, we expanded and hopefully made more clear how we conceptualize the problem, use the data, andestablish our method. In Figure 2, we enriched the y labels of each panel with the name associated with the order parameter.

(ii) We thank the reviewer for helping us improve the readability of the introductory part, thus providing moreinsights into disordered systems techniques for a broader audience. We have added a few explanations at the end of page 2 – to explain the advantages of such methodology compared to other strategies and models.

(iii) We thank the reviewer for raising the need for a more in-depth ecological discussion of our results. A simple wayto understand why neutral effects may dominate in healthy patients is the following. Neutrality implies that species differences are mainly shaped by stochastic processes such as demographic noise, with species treated as different realizations of the same underlying stochastic ecological dynamics. In our analysis, we observe that healthy individuals tend to exhibit highly similar microbial communities, suggesting that the compositional variability among their microbiomes is compatible—at least in part—with the fluctuations expected from demographic stochasticity alone. In contrast, patients with the disease display significantly more heterogeneous microbial compositions. The diversity and structure of their gut communities cannot be satisfactorily explained by neutral demographic fluctuations alone.

This discrepancy implies that additional deterministic forces—such as altered ecological interactions—are driving the divergence observed in dysbiotic states. In diseased individuals, the breakdown of such interactions leads to a structurally distinct regime that may correspond to a phase of marginal stability, as indicated by our theoretical modeling. This shift marks a transition from a community governed by neutrality and demographic noise to one dominated by non-neutral ecological forces (as depicted in Figure 4). We added these comments in the discussion section of the revised manuscript.

(2) Taking into account the broader audience, we invite the authors to edit the abstract, as it seems to jump from one ecological concept to another without explicitly communicating what is the link between these concepts. From the first two sentences, the motivation seems to be species diversity, but no mention of diversity comes after the second sentence. There is no proper introduction/definition of what macroecological states are. After that, the authors switch to healthy and unhealthy states, without previously introducing any link between gut microbiota states and the host’s health (which perhaps could be good in the first or second sentence, although other framings can be as valid). After that, interactions appear in the text and are related to instability, but the reader might not know whether this is surprising or if healthy/unhealthy states are generally related to stability.We pointed out a few examples, but the authors could extend their revision on (i), (ii) and (iii) beyond such specific comments. In our opinion, this would really benefit the paper.

In response to the reviewer’s concern about conceptual clarity and structure, we substantially revised the abstract to improve its accessibility and logical flow. In the revised abstract, we now clearly link species diversity to microbiome structure and function from the outset, addressing initial confusion. We provide a concise definition of ”macroecological states,” framing them as reproducible statistical patterns reflecting community-level properties. Additionally, the revised version explicitly connects gut microbiome states to host health earlier, resolving the previous abrupt shift in focus. Finally, we conclude by highlighting how disordered systems theory advances our understanding of microbiome stability and functioning, reinforcing the novelty and broader significance of our approach. Overall, the revised abstract better serves a broad interdisciplinary audience, including readers unfamiliar with the technicalities of disordered systems or microbial ecology, while preserving the scientific depth and accuracy of our work

(3) The connection with consumer-resource (CR) models is quite unusual. In Equation (12), why do the authors assume that the consumption term does not depend on R? This should be addressed, since this term is usually dependent on R in microbial ecology models.In case this is helpful, it is known that the symmetric Lotka-Volterra model emerges from time-scale separation in the MacArthur model, where resources reproduce logistically and are consumed by other species (e.g., plants eaten by herbivores). Consumer-resource models form a broad category, while the MacArthur model is a specific case featuring logistic resource growth. For microbes, a more meaningful justification of the generalized Lotka-Volterra (GLV) model from a consumer-resource perspective involves the consumer-resource dynamics in a chemostat, where time-scale separation is assumed and higher-order interactions are neglected. See, for example: (a) The classic paper by MacArthur: R. MacArthur. Species packing and competitive equilibrium for many species. Theoretical Population Biology, 1(1):1-11, 1970. (b) Recent works on time-scale separation in chemostat consumer-resource models: Anna Posfai et al., PRL, 2017 Sireci et al., PNAS, 2023 Akshit Goyal et al., PRX-Life, 2025

We thank the reviewer for the observation. We apologize for the typo that appeared in the main text and that we promptly corrected. The Consumers-Resources model we had in mind is the classical case proposed by MacArthur, where resources are self-regulated according to a logistic growth mechanism, which leads to the generalized LotkaVolterra model we employ in our work.

Minor concerns:(1) The title has a nice pun for statistical physicists, but we wonder if it can be a bit confusing for the broader audience of e-Life. Although we leave this to the author’s decision, we’d recommend considering changing the title, making it more explicit in communicating the main contribution/result of the work.

Following the reviewer’s suggestion, we have introduced an explanatory subtitle: “Linking Species Interactions to Dysbiosis through a Disordered Lotka-Volterra Framework”.

(2) Review the references - some preprints might have already been published: Pasqualini J. 2023, Sireci 2022, Wu 2021.

We thank the reviewer for pointing our attention to this inaccuracy. We updated the references to Pasqualini and Sireci papers. To our knowledge, Wu’s paper has appeared as an arXiv preprint only.

(3) Species do not generally exhibit identical carrying capacities see Grilli, Nat. Commun., 2020; some taxa are generally more abundant than others. The authors could discuss whether the model, with the inferred parameters, can accurately reproduce the distribution of species’ mean abundances.

We thank the reviewer for this insightful comment. As discussed in the revised manuscript (lines 294–299), our current model does not accurately reproduce the empirical species abundance distribution (SAD). This limitation stems from the assumption of constant carrying capacities across species. While empirical observations (e.g., Grilli et al., Nat. Commun., 2020 [1]) show heterogeneous mean abundances often following power-law or log-normal distributions. However, our model assumes constant carrying capacity, resulting in SADs devoid of fat tails, which diverge from empirical data.

This simplification is implemented to maintain the analytical tractability of the disordered generalized Lotka-Volterra (dGLV) framework, a common approach also found in prior works such as Bunin (2017) and Barbier et al. (2018) [2, 3]. Introducing heterogeneity in carrying capacities, such as drawing them from a log-normal distribution, or switching to multiplicative (rather than demographic) noise, could indeed produce SADs that better align with empirical data. Nevertheless, implementing changes would significantly complicate the analytical treatment.

We acknowledge these directions as promising avenues for future research. They could help enhance the empirical realism of the model and its capacity to capture observed macroecological patterns while posing new theoretical challenges for disordered systems analysis

(4) A substantial number of cited works (Grilli, Nat. Commun., 2020; Zaoli & Grilli, Science Advances, 2021; Sireci et al., PNAS, 2023; Po-Yi Ho et al., eLife, 2022) suggest that environmental fluctuations play a crucial role in shaping microbiome composition and dynamics. Is the authors’ analysis consistent with this perspective? Do they expect their conclusions to remain robust if environmental fluctuations are introduced?

We thank the reviewer for stressing this point. The introduction of environmental fluctuations in the model formally violates detailed balance, thereby preventing the definition of an energy function. To date, no study has integrated random interactions together with both demographic and environmental noise within a unified analytical framework. This is certainly a highly promising direction that some of the authors are already exploring. However, given the inherently out-of-equilibrium nature of the system and the absence of a free energy, we would need to adopt a Dynamical Mean-Field Theory formalism and eventually analyze the corresponding stationary equations to be solved self-consistently. We added, however, a brief note in the Discussion section.

(5) The term “order parameters“ may not be intuitive for a biological audience. In any case, the authors should explicitly define each order parameter when first introduced.

We thank the reviewer for the comment. We introduced the names of the order parameters as soon as they are introduced, along with a brief explanation of their meaning that may be accessible to an audience with biological background.

(6) Line 242: Should ψU be ψD?

We thank the reviewer for the observation. We corrected the typo.

(7) Given that the authors are discussing healthy and diseased states and to avoid confusion, the authors could perhaps use another word for ’pathological’ when they refer to dynamical regimes (e.g., in Appendix 2: ’letting the system enter the pathological regime of unbounded growth’).

We thank the reviewer for the helpful comment. As suggested, we used the term “unphysical” instead of “pathological” where needed.

**Reviewer #2:**
(1) A technical point that I could not understand is how the authors deal with compositional data. One reason for my confusion is that the order parameters h and q0 are fixed n data to 1/S and 1/S2, and thus I do not see how they can be informative. Same for carrying capacity, why is it not 1 if considering relative abundance?

We thank the reviewer for raising this point. We acknowledge that the treatment of compositional data and the interpretation of order parameters h and q0 were not sufficiently clarified in the manuscript. Additionally, there was an imprecision in the text regarding the interpretation of these parameters.

As defined in revised Eq. (4) of the manuscript, h and q0 are to be averaged over the entire dataset, summing across samples α. Specifically, \begin{document}$h={\overline{{1 / S_{\alpha}}} } $\end{document} and \begin{document}$q_{0}={\overline{{1 / S_{\alpha}^{2}}} } $\end{document}, where S_α_ is the number of species present in sample α and is the average over samples. These parameters are therefore informative, as they encapsulate sample-level ecological diversity, and their variation reflects biological differences between healthy and diseased states. For instance, Pasqualini et al., 2024 [4] reported significant differences in these metrics between health conditions, thereby supporting their ecological relevance.

Regarding carrying capacities, we clarify that although we work with relative abundance data (i.e., compositional data), we do not fix the carrying capacity K to 1. Instead, we set K to the maximum value of xi (relative abundance) within each sample, to preserve compatibility with empirical data and allow for coexistence. While this remains a modeling assumption, it ensures better ecological realism within the constraints of the disordered GLV framework.

(2) Obviously I’m missing something, so it would be nice to clarify in simple terms the logic of the argument. I understand that Lagrange multipliers are going to be used in the model analysis, and there are a lot of technical arguments presented in the paper, but I would like a much more intuitive explanation about the way the data can be used to infer order parameters if those are fixed by definition in compositional data.

We thank the reviewer for the observation. The order parameters can be measured directly from the data, even in the presence of compositionality, as explained above. We can connect those parameters with the theory even for compositional data, because the only effect of adding the compositionality constraint is to shift the linear coefficient in the Hamiltonian, which corresponds to shifting the average interaction µ. However, the resulting phase diagram is mostly affected by the variance of the interactions σ2 (as µ is such that we are in the bounded phase).

(3) Another point that I did not understand comes from the fact that the authors claim that interaction variance is smaller in unhealthy microbiomes. Yet they also find that those are closer to instability, and are more driven by niche processes. I would have expected the opposite to be true, more variance in the interactions leading to instability (as in May’s original paper for instance). Is this apparent paradox explained by covariations in demographic stochasticity (T) and immigration rate (lambda)? If so, I think it would be very useful to comment on that.

As Altieri and coworkers showed in their PRL (2021) [5], the phase diagram of our model differs fundamentally from that of Biroli et al. (2018) [6]. In the latter, the intuitive rule – greater interaction variance yields greater instability – indeed holds. For the sake of clarity, we have attached below the resulting phase diagram obtained by Altieri et al.

The apparent paradox arises because the two phase diagrams are tuned by different parameters. Consequently, even at low temperature and with weak interaction variance, our system may sit nearer to the replica-symmetrybreaking (RSB) line.

Fig. 3 in the main text it is not a (σ,T) phase diagram where all other parameters are kept constant. Rather, it is a plot of the inferred σ and T parameters from the data (without showing the corresponding µ).

To capture the full, non-trivial influence of all parameters on stability, we studied the so-called “replicon eigenvalue” in the RS (i.e. single equilibrium) approximation. This leading eigenvalue measures how close a given set of inferred parameters – and hence a microbiome – is to the RSB threshold. For a visual representation of these findings, refer to Figure 4.

**Author response image 1. sa4fig1:** 

(4) What do the empirical SAD look like? It would be nice to see the actual data and how the theoretical SADs compare.

The empirical species abundance distributions (SADs) analyzed in our study are presented and discussed in detail in Pasqualini et al., 2024 [4]. Given the overlap in content, we chose not to reproduce these figures in the current manuscript to avoid redundancy.

As we also clarify in the revised text, the theoretical SAD is derived from the disordered generalized Lotka-Volterra (dGLV) model in the unique fixed point phase typically exhibit exponential tails. These distributions do not match the heavier-tailed patterns (e.g., log-normal or power-law-like) observed in empirical microbiome data. This discrepancy stems from the simplifying assumptions of the dGLV framework, including the use of constant carrying capacities and demographic noise.

In the revised manuscript, we have added a brief discussion in the revised manuscript to explicitly acknowledge this limitation and emphasize it as a direction for future refinement of the model, such as incorporating heterogeneous carrying capacities or exploring alternative noise structures.

(5) Some typos: often “niche” is written “nice”.

We thank the reviewer for this suggestion. After inspecting the text, we corrected the reported typos.

**Reviewer #3:**
Major comments:(1) In the S3 text, the authors say that filtered metagenomic reads were processed using the software Kaiju. The description of the pipeline does not mention how core genes were selected, which is often a crucial step in determining the abundance of a species in a metagenomic sample. In addition, the senior author of this manuscript has published a version of Kaiju that leverages marker genes classification methods (deemed Core-Kaiju), but it was not used for either this manuscript or Pasqualini et al. (2014; Tovo et al., 2020). I am not suggesting that the data necessarily needs to be reprocessed, but it would be useful to know how core genes were chosen in Pasqualini et al. and why Core-Kaiju was not used (2014).

Prior to the current manuscript and the PLOS Computational Biology paper by Pasqualini et al. [4], we applied the core-Kaiju protocol to the same dataset used in both studies. However, this tool was originally developed and validated using general catalogs of culturable organisms, not specifically tuned for gut microbiomes. As a result, we have realized that in many samples Core Kajiu would filter only very few species (in some samples, the number of identified species was as low as 5–10), undermining the reliability of the analysis. Due to these limitations, we opted to use the standard Kaiju version in our work. We are actively developing an improved version of the core-Kaiju protocol that will overcome the discussed limitations and preliminary results (not shown here) indicate the robustness of the obtained patterns also in this case.

(2) My understanding of Pasqualini et al. was that diseased patients experienced larger fluctuations in abundance, while in this study, they had smaller fluctuations (Figure 3a; 2024). Is this a discrepancy between the two models or is there a more nuanced interpretation?

We thank the reviewer for the observation. This is only an apparent discrepancy, as the term fluctuation has different meanings in the two contexts. The fluctuations referred to by the reviewer correspond to a parameter of our theory—namely, noise in the interactions. Conversely, in Pasqualini et al. σ indicates environmental fluctuations. Nevertheless, there is no conceptual discrepancy in our results: in both studies, unhealthy microbiomes were found to be less stable. In fact, also in this study, notably Fig. 4, shows that unhealthy microbiomes lie closer to the RSB line, a phenomenon that is also associated with enhanced fluctuations.

(3) Line 38-41: It would be helpful to explicitly state what “interaction patterns” are being referenced here. The final sentence could also be clarified. Do microbiomes “host“ interactions or are they better described as a property (“have”, “harbor”). The word “host” may confuse some readers since it is often used to refer to the human host. I am also not sure what point is being made by “expected to govern natural ones”. There are interactions between members of a microbiome; experimental studies have characterized some of these interactions, which we expect to relate in some way to interactions in nature. Is this what the authors are saying?

Thanks. We agree that this sentence was not clear. Indeed, we are referring to pairwise species interactions and not to host-microbiome interactions. We have rewritten this part in the following way: In fact, recent work shows that the network-level properties of species-species interactions —for example, the sign balance, average strength, and connectivity of the inferred interaction matrix— shift systematically between healthy and dysbiotic gut communities (see for instance, [7, 8]). Pairwise species interactions have been quantified in simplified in-vitro consortia [9, 10]; we assume that the same classes of interactions also operate—albeit in a more complex form—in the native gut microbiome.

(4) Line 43: I appreciate that the authors separated neutral vs. logistic models here.(5) Lines 51-75: The framing here is well-written and convincing. Network inference is an ongoing, active subject in ecology, and there is an unfortunate focus on inferring every individual interaction because ecologists with biology backgrounds are not trained to think about the problem in the language of statistical physics.

We thank the reviewer for these positive comments.

(6) Line 87: Perhaps I’m missing something obvious, but I don’t see how ρi sets the intrinsic timescale of the dynamics when its units are 1/(time*individuals), assuming the dimensions of ri are inverse time.

We thank the reviewer for the observation. We corrected this phrase in the main text.

(7) Lines 189-190: “as close as possible to the data” it would aid the reader if you specified the criteria meant by this statement.

We thank the reviewer for the observation. We removed the sentence, as it introduced some redundancy in our argument. In the subsequent text, the proposed method is exposed in details.

(8) Line 198: It would aid the reader if you provided some context for what the T - σ plane represents.

We thank the referee for the helpful indication. Indeed, we have better clarified the mutual role of the demographic noise amplitude and strength of the random interaction matrix, as theoretically predicted in the PRL (2021) by Altieri and coworkers [5]. Please, find an additional paragraph on page 6 of the resubmitted version.

(9) Line 217: Specifying what is meant by “internal modes“ would aid the typical life science reader.

We thank the reviewer for the suggestion. Recognizing that referring to “internal modes” to describe the SAD shape in that context might cause confusion, we replaced “internal modes“ with “peaks”.

(10) Line 219: Some additional justification and clarification are needed here, as some may think of “m“ as being biomass.

We added a sentence to better explain this concept. “In classical and quantum field theory, the particle-particle interaction embedded in the quadratic term is typically referred to as a mass source. In the context of this study, \begin{document}$m=1-\beta \sigma^{2}\left(q_{d}-q_{0}\right)$\end{document} captures quadratic fluctuations of species abundances, as also appearing in the expression of the leading eigenvalue of the stability matrix.”

Minor comments:(1) I commend the authors for removing metagenomic reads that mapped to the human genome in the preprocessing stage of their pipeline. This may seem like an obvious pre-processing step, but it is unfortunately not always implemented.

We thank the referee for pointing this potential issue. The data used in this work, as well as the bioinformatic workflow used to generate them has been described in detail in Pasqualini et al., 2024 [4]. As one of the main steps for preprocessing, we remove reads mapping to the human genome.

(2) Line 13: “Bacterial“ excludes archaea, and while you may not have many high-abundance archaea in your human gut data, this sentence does not specify the human gut. Usually, this exclusion is averted via the term “microbial“, though sometimes researchers raise objections to the term when the data does not include fungal members (e.g., all 16S studies).

We thank the reviewer for this suggestion. As to include archaeal organisms, we adopt the term “microbial“ instead of “bacterial“.

(3) Line 18: This manuscript is being submitted under the “Physics of Living Systems“ tract, but it may be useful to explicitly state in the Abstract that disordered systems are a useful approach for understanding large, complex communities for the benefit of life science researchers coming from a biology background.

Thank. We have modified the abstract following this suggestion.

(4) Line 68: Consider using “adapted“ or something similar instead of “mutated“ if there is no specific reason for that word choice.

We thank the reviewer for this suggestion, which was implemented in the text.

(5) Line 111: It would be useful to define annealed and quenched for a general life science audience.

We thank the reviewer for this suggestion. In the “Results” section, we have opted for “time-dependent disordered interactions” to reach a broader audience and avoid any jargon. Moreover, in the Discussion we added a detailed footnote: “In contrast to the quenched approximation, the annealed version assumes that the random couplings are not fixed but instead fluctuate over time, with their covariance governed by independent Ornstein–Uhlenbeck processes.”

(6) Line 124: Likewise for the replicon sector.

We thank the reviewer for the suggestion. We added a footnote on page 4, after the formula, to highlight the physical intuition behind the introduction of the replicon mode.

“The replicon eigenvalue refers to a particular type of fluctuation around the saddle-point (mean-field) solution within the replica framework. When the Hessian matrix of the replicated free energy is diagonalized, fluctuations are divided into three sectors: longitudinal, anomalous, and replicon. The replicon mode is the most sensitive to criticality signaling – by its vanishing trend – the emergence of many nearly-degenerate states. It essentially describes how ‘soft’ the system is to microscopic rearrangements in configuration space.”

(7) Figure 2: It would be helpful to include y-axis labels for each order parameter alongside the mathematical notation.

We thank the reviewer for this suggestion. Now the y-axis of Figure 2 includes, along the mathmetical symbol, the label of the represented quantities.

(8) Line 242: Subscript “U” is used to denote “Unhealthy” microbiomes, but “D” is used to denote “Diseased” in Figs. 2 and 3 (perhaps elsewhere as well).

We thank the reviewer for this observation. After checking the various subscripts in the text, coherently with figure 2 and 3, we homogenized our notation, adopting the subscript “D“ for symbols related to the diseased/unhealthy condition.

(9) Line 283: “not to“ should be “not due to“

We thank the reviewer for this suggestion. After inspecting the text, we corrected the reported error.

(10) Equations 23, 34: Extra “=“ on the RHS of the first line.

We consistently follow the same formatting across all the line breaks in the equations throughout the text.

We are thus resubmitting our paper, hoping to have satisfactorily addressed all referees’ concerns.

**References**

(1) Jacopo Grilli. Macroecological laws describe variation and diversity in microbial communities. Nature communications, 11(1):4743, 2020.

(2) Guy Bunin. Ecological communities with lotka-volterra dynamics. Physical Review E, 95(4):042414, 2017.

(3) Matthieu Barbier, Jean-Franc¸ois Arnoldi, Guy Bunin, and Michel Loreau. Generic assembly patterns in complex ecological communities. Proceedings of the National Academy of Sciences, 115(9):2156–2161, 2018.

(4) Jacopo Pasqualini, Sonia Facchin, Andrea Rinaldo, Amos Maritan, Edoardo Savarino, and Samir Suweis. Emergent ecological patterns and modelling of gut microbiomes in health and in disease. PLOS Computational Biology, 20(9):e1012482, 2024.

(5) Ada Altieri, Felix Roy, Chiara Cammarota, and Giulio Biroli. Properties of equilibria and glassy phases of the random lotka-volterra model with demographic noise. Physical Review Letters, 126(25):258301, 2021.

(6) Giulio Biroli, Guy Bunin, and Chiara Cammarota. Marginally stable equilibria in critical ecosystems. New Journal of Physics, 20(8):083051, 2018.

(7) Amir Bashan, Travis E Gibson, Jonathan Friedman, Vincent J Carey, Scott T Weiss, Elizabeth L Hohmann, and Yang-Yu Liu. Universality of human microbial dynamics. Nature, 534(7606):259–262, 2016.

(8) Marcello Seppi, Jacopo Pasqualini, Sonia Facchin, Edoardo Vincenzo Savarino, and Samir Suweis. Emergent functional organization of gut microbiomes in health and diseases. Biomolecules, 14(1):5, 2023.

(9) Jared Kehe, Anthony Ortiz, Anthony Kulesa, Jeff Gore, Paul C Blainey, and Jonathan Friedman. Positive interactions are common among culturable bacteria. Science advances, 7(45):eabi7159, 2021.

(10) Ophelia S Venturelli, Alex V Carr, Garth Fisher, Ryan H Hsu, Rebecca Lau, Benjamin P Bowen, Susan Hromada, Trent Northen, and Adam P Arkin. Deciphering microbial interactions in synthetic human gut microbiome communities. Molecular systems biology, 14(6):e8157, 2018.